# THE DECRYPTO BENCHMARK FOR MULTI-AGENT REASONING AND THEORY OF MIND

## ABSTRACT

As Large Language Models (LLMs) gain agentic abilities, they will have to navigate complex multi-agent scenarios, interacting with human users and other agents in cooperative and competitive settings. This will require new reasoning skills, a crucial one being *theory of mind* (ToM), or the ability to reason about the "mental" states of other agents. However, ToM and other multi-agent abilities in LLMs are poorly understood, since existing benchmarks suffer from narrow scope, data leakage, saturation, and lack of interactivity. We thus propose DECRYPTO, a game-based benchmark for multi-agent reasoning and ToM drawing inspiration from cognitive science, computational pragmatics and multi-agent reinforcement learning. It is designed to be as *easy* as possible in all other dimensions, eliminating confounding factors commonly found in other benchmarks. To our knowledge, it is also the first platform for designing interactive ToM experiments.

We validate the benchmark design through comprehensive empirical evaluations of frontier LLMs, robustness studies, and human-AI cross-play experiments. We find that LLM game-playing abilities lag behind humans and simple word-embedding baselines. We then create variants of two classic cognitive science experiments within DECRYPTO to evaluate three key ToM abilities. Surprisingly, we find that state-of-the-art reasoning models are significantly *worse* at those tasks than their older counterparts. This demonstrates that DECRYPTO addresses a crucial gap in current reasoning and ToM evaluations, and paves the path towards better artificial agents. Code at `anonymous.4open.science/r/decrypto/`.

## 1 INTRODUCTION

Reasoning abilities and agentic behaviour are currently at the forefront of large language model (LLM) research (Huang et al., 2023; Jaech et al., 2024; Guo et al., 2025), with early applications of LLM agents including software engineering (Jimenez et al., 2023) and web navigation (Yao et al., 2023). While those tasks are predominantly single-agent, real-world agentic applications will increasingly involve complex interactions with many other actors, including business partners, teammates, users and other agents. Thus, the future of AI problems is irreducibly *multi-step*, *multi-agent*, *partially observable*, and *stochastic* – a reality not captured by the existing suite of benchmarks.

To navigate these multi-agent scenarios, LLMs will require *theory of mind* (ToM), the general ability of modelling the mental states of other agents (artificial or biological), including their knowledge, beliefs, intentions and abilities. However, LLM ToM remains poorly understood. Despite ToM being inherently tied to multi-agent *interactions*, the majority of existing benchmarks lack interactivity and are restricted to variants of the Sally-Anne task (Baron-Cohen et al., 1985). Furthermore, Hu et al. argue that many of those benchmarks suffer from confounding factors and bias introduced when translating embodied scenarios to text.

To address the gap in ToM evaluations, we introduce DECRYPTO, a multi-agent reasoning and ToM benchmark based on the award-winning language game of the same name. DECRYPTO is a pragmatic inference game (Degen, 2023), where two agents must exchange secret messages while preventing a third party from intercepting them.

We designed DECRYPTO to be future-proof and as *easy* as possible for LLMs in all other dimensions except multi-agent reasoning. Indeed, it is operated entirely through word associations, eliminating common confounding factors of LLM performance, such as tokenisation, long contexts or embodied scenarios. The benchmark enables the study of cooperation, competition and ToM within a single game, and requires players to reason about the information available to others before making strate-

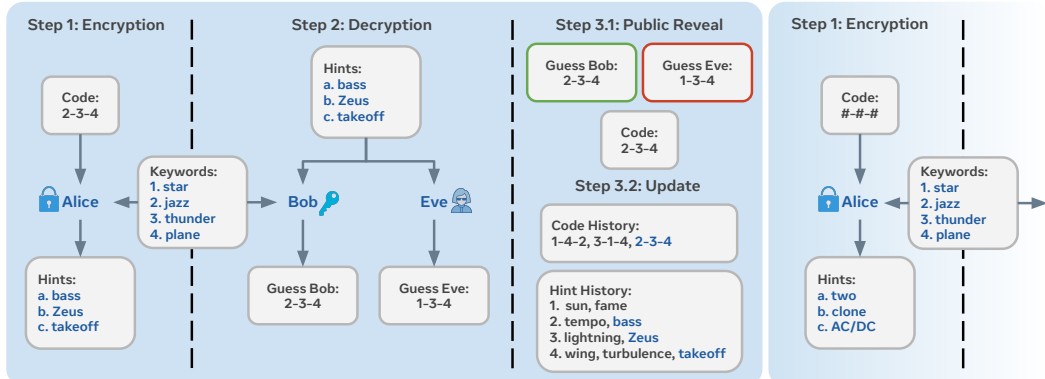

Figure 1: Left: Overview of a turn in DECRYPTO, split into three steps, with Alice and Bob (Encoder and Decoder) playing against Eve (Interceptor). Step 1: Alice draws a code of 3 non-repeating digits, and provides 3 hints referring to the meaning of the four *Keywords*. Step 2: Bob and Eve receive the hints and attempt to guess the code independently. Step 3: Both guesses and the code are publicly revealed, and the code and hint histories are updated. All players have access to the histories at all times, but only Alice and Bob have access to the keywords. Alice aims to give hints that are easy for Bob to decode but hard for Eve. As the game progresses, the growing hint history makes it easier for Eve to intercept. Right: Encryption phase of a new turn. **We invite the reader to guess the code as Bob and then copy this for the answer: [        ].** We provide a rationale in Section C.

gic decisions. Since it is based on a popular boardgame, our benchmark produces easily interpretable insights and enables the study of human-AI interactions in strategic communication tasks. Finally, to the best of our knowledge, DECRYPTO is the first platform for studying LLM ToM in interactive settings. Built to be easily extended, it enables researchers to create new ToM experiments by writing only prompts and a few lines of code. We demonstrate the versatility of the DECRYPTO ToM platform by creating variants of two classic experiments from cognitive psychology.

## 2 DECRYPTO: THE GAME

DECRYPTO is a word-based code-guessing board game published by Scorpion Masqué[1], of which we implement a variant requiring three players divided into two teams. Borrowing the terminology from cryptography, players are assigned three distinct roles: Alice (the Encoder), Bob (the Decoder), and Eve (the Interceptor or Eavesdropper). A full game (i.e., an episode) lasts at most 8 turns. We show one such turn in Figure 1.

Alice and Bob are on the same team, and share 4 ordered secret *keywords*, sampled without replacement before the start of the episode and kept fixed throughout. In Figure 1, those are [star, jazz, thunder, plane].

Each turn, Alice samples a secret *code* of 3 non-repeating digits between 1 and 4, such as [2-3-4], with any of the 24 possible codes occurring at most once per episode. Alice must then provide 3 public *hints* (e.g. [bass, Zeus, takeoff], such that Bob can guess the code but Eve cannot.

Next, Bob and Eve each make an independent attempt to *guess* the code. If Bob guesses incorrectly, his team gets a *Miscommunication* token. If Eve guesses correctly, she gets an *Intercept* token.

Finally, Bob and Eve's guesses, the real code, and any event (miscommunication or intercept) that occurred during the turn are publicly revealed. Thus, the history of all codes from previous turns and the history of past hints associated with each of the four keywords are common knowledge, but the keywords themselves are not. The ***code history*** and especially the ***hint history*** are the two main pieces of information that will help Eve make educated guesses in future turns. For instance, if the hint history for digit 3 so far is [lightning, Zeus], Eve may associate terms related to storms or electricity with that digit, even without knowing the exact keyword.

The game ends and Eve wins immediately if Alice and Bob accumulate two Miscommunication tokens or if Eve gains two Intercept tokens. Alice and Bob win if they make it through 8 rounds

---
[1]www.scorpionmasque.com/en/decrypto

without either of those two conditions happening. Alice must therefore be careful to provide subtle hints to avoid interception, yet make them clear enough for Bob to guess correctly.

The keywords are sampled from a predetermined set, but *the choice of hints is open-ended* and can be composed of multiple words. The restriction is that hints must be real words (including proper nouns) and refer to the meaning of the keywords, not to their spelling or pronunciation. They must also rely on publicly available information, which can include culture, science or recent events.

**Beyond word associations.** At the surface level, DECRYPTO provides a language reasoning challenge that consists in matching hints to either keywords or the hint history. However, Alice's hints cannot be too literal, or they will get intercepted, and so DECRYPTO can be formalised as a pragmatic inference game under the Rational Speech Act (RSA) framework (Goodman & Frank, 2016; Degen, 2023), where the listeners (Bob and Eve) update their belief of the intended meaning of the speaker (Alice) via Bayesian inference. We provide such a treatment in Section H, explicitly showing that agents must model each other's beliefs and perform second-order ToM for optimal play.

To help readers intuit the role of ToM in Decrypto, we invite them to attempt guessing the code on the right side of Figure 1 as Bob, and to read the rationale in Section C. Alternatively, readers can download our codebase and play a game of Decrypto with an LLM in only a few minutes.

## 3 DECRYPTO: THE BENCHMARK

**An easy benchmark.** Unlike other reasoning benchmarks, DECRYPTO is purposely designed to *not* require many of the capabilities that large pre-trained models struggle with. In particular, it demands no symbolic (Bard et al., 2020), mathematical (Cobbe et al., 2021; Zhang et al., 2024a) or spatial reasoning (Clark et al., 2018; Carroll et al., 2019), no tool use (Xu et al., 2023a), and no particular attention to tokenisation, which has been shown to affect arithmetic performance (Singh & Strouse, 2024). It also distinguishes itself from recent ToM benchmarks (Le et al., 2019; He et al., 2023; Kim et al., 2023; Zhou et al., 2023; Xu et al., 2024a; Gandhi et al., 2024; Sclar et al., 2024) by not relying on state tracking (e.g. of object locations) or on textual representations of embodied settings, which Hu et al. argue introduce unwanted pragmatic artifacts and bias LLM answers.

Instead, strong performance in DECRYPTO relies on language-based reasoning and theory of mind. Since LLMs are trained to learn word co-occurrences and follow instructions, we expect DECRYPTO to be easy to play by LLMs, but hard to master. Our results and Section L show this to be the case: even Llama 3.1-8B understands the rules, but much stronger models still struggle to play well.

**A *future-proof* benchmark.** With its two-team, three-player dynamics, DECRYPTO is a benchmark for both competitive and cooperative scenarios.

The multi-agent and open-ended nature of the game means that difficulty scales with the choice of agents for each role, resulting in an evergreen challenge that is much harder to saturate than a fixed dataset of problems. We curate a set of 680 possible keywords, resulting in over 8.8 billion possible keyword combinations to limit memorisation, and also implement the option to provide custom keywords beyond this standard set. Combined with the option to provide arbitrary hints, including ones that refer to recent events or niche cultural works, the number of possible games is virtually infinite. Furthermore, DECRYPTO was originally designed for human play, facilitating studies on human-AI coordination and comparisons in addition to inter-AI benchmarking.

**Competition.** In this setting, we compare two models $A$ and $B$ by having both Alice and Bob be played by $A$, and Eve by $B$. At its simplest, we can record the win rate of $A$ vs. $B$, swapping teams to ensure a fair evaluation. However, we propose more granular metrics below.

**Ad-hoc Coordination.** Here, we fix Eve to a give model and instantiate Alice and Bob to be played by different models $A$ and $B$. Like in Stone et al. (2010), we are concerned with the ability to "*efficiently and robustly collaborate with previously unknown teammates*", such as independently trained LLMs. As highlighted in FAIR et al. (2022), this requires approaches beyond simple self-play, since the goal is to adopt strategies that *generalize* to team pairings where $A \neq B$. A crucial subset of ad-hoc coordination is **human-AI coordination**, where one of the two agents (Alice or Bob) is played by a human. This setting paves the way towards more social AI agents that seamlessly coordinate with humans and understand their intents.

**Metrics.**    Both settings are subject to the same tension that is at the core of DECRYPTO: Alice must provide hints that balance what she knows about Bob, Eve, and the information available to each of them. If the hints are too obscure, Bob will guess wrong, which leads to a ***miscommunication***; too obvious, and Eve will ***intercept***; just right, and Alice and Bob survive for another round. The number of miscommunications and of intercepts are therefore two sides of the same coin, providing granular break-down of the failure modes of LLMs. Meanwhile, the average *number of turns per episode* captures both sides in one metric, since longer games mean that Alice and Bob could better balance the difficulty of hints to avoid defeat. Game length is also more informative than win-rate, since we empirically find current LLMs to be much weaker at providing hints than intercepting, which results in Eve having significantly higher win-rates in most match-ups (see Figure 7).

## 3.1 Theory of Mind

As explained in Section 2, strong performance in DECRYPTO requires players to explicitly reason about each other's beliefs and world knowledge. Indeed, a model could gain significant advantage over its opponent by identifying and exploiting their knowledge gaps, either due to human limitations or by referring to information made public after the knowledge cut-off of an older model.

Beyond game playing, DECRYPTO provides a platform for conducting interactive ToM experiments inspired by cognitive psychology. We conduct two such experiments, evaluating three different ToM abilities, and gaining diagnostic insight into the poor performance of LLMs in DECRYPTO. Thanks to a flexible codebase design, each required only new prompts and approximately 15 additional lines of code. We describe these ToM experiments and their results in Section 5.1.

## 4 Specialist vs Generalist Agents

Given the plurality of settings in DECRYPTO, we caution users to be explicit and transparent about their assumptions when declaring state-of-the-art. When reporting results, we recommend distinguishing between two classes of agents: **generalists** and **specialists**.

**Generalist agents** refer to general-purpose models used to play the game "out-of-the-box", without any additional fine-tuning or in-context learning outside of that happening within individual games. In particular, for generalist agents, we *consider the prompt to be the observation* returned by the environment, which precludes any form of prompt engineering. Evaluating foundation models in this framework assesses their ability to zero-shot generalise to novel tasks, including out-of-distribution ones and tasks where training data is limited or unavailable.

Because foundation models are often sensitive to prompting, we recommend validating generalist agents with a range of different prompts to prevent the choice of prompt favouring any one model. We provide such an evaluation using 625 prompt variants in Section G, showing that game performance is robust to prompt variations and depends predominantly on general model capabilities.

In contrast, **specialist agents** are all the agents designed specifically for the task. Solutions in this space include rule-based strategies, training on the game using RL, building a prompt pipeline informed by the rules of the games, and prompt engineering. Here, *the environment only returns key information*, and prompt engineering is the designer's burden, if it is needed at all.

Specialist agents can, in theory, overfit to DECRYPTO, which makes some degree of performance easier to reach, but does *not* automatically solve the game under every setting. For instance, we know from the cooperative multi-agent RL literature that training models to collaborate in cross-play (XP) with unseen agents can be extremely challenging, even with unlimited compute and access to the environment (Hu et al., 2021; Cui et al., 2023; FAIR et al., 2022). We expect this to also hold for DECRYPTO. This is evidenced by the rule-based baselines we describe next, which coordinate flawlessly in DECRYPTO if Alice and Bob pre-agree on a common strategy and a word embeddings, but otherwise can fail catastrophically when paired with other agents, including humans and LLMs.

**Specialist Agents: Baselines.**    We introduce two specialist agent baselines to play Decrypto, built around word embedding models. Those are the common crawl GloVe with a vocabulary of 2.2 million words (Pennington et al., 2014) and the Word2Vec embeddings trained on the Google News dataset, provided by the gensim library (Rehurek & Sojka, 2011; Mikolov, 2013).

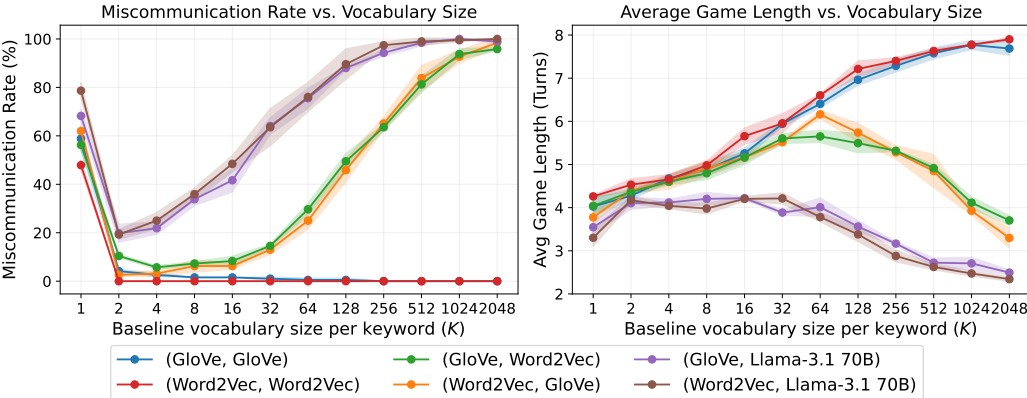

Figure 2: Percentage of games ending in miscommunications (left) and average game length (right) for the word embedding baselines as a function of $K$ – the hint vocabulary size per keyword. Each curve corresponds to an encoder-decoder pair. For low $K$, the baselines coordinate well, but their hints are easier to intercept. For high $K$, both models sample from lower similarity words. This results in very strong self-play baselines, which cannot coordinate in cross-play, with most games ending in miscommunication for $K > 128$. While the interceptor is Llama 3.1-70B, trends hold across all interceptors, including baselines. We report mean and standard error over 3 model seeds.

For both models, the baseline algorithm is the same and relies on a fixed corpus of 5696 nouns. When hinting a digit, Alice draws from the set of the top-$K$ most similar words to the corresponding keyword under cosine similarity, filtering out hints that are more similar to another keyword than to the one intended. In turn, Bob greedily assigns each hint to the most similar keyword, measured using his own embedding model and cosine similarity. Not having access to the keywords, Eve computes the mean embedding of the hint history for each keyword, and then finds the assignment between hints and mean embedding that maximises the sum of cosine similarities between each hint and the history of the digit it is assigned to. We provide additional algorithmic details in Section D.

The purpose of these baselines is threefold. First, we can study the case where Alice and Bob share perfect ToM, represented by a shared strategy and word embedding. This establishes an upper bound on performance to which to compare other agents. Second, having baselines using two different word embeddings, we demonstrate in Section 5 how unaccounted differences in how agents associate words can lead to a catastrophic lack of coordination and poor XP performance. Third, $K$ determines the amount of diversity in Alice's hints. By tweaking it, we can operate in a regime where GloVe and Word2Vec correlate and represent reasonable rule-based baselines against which we can compare LLMs. We thus use $K = 16$ for the rest of the paper, as it is the value in Figure 2 beyond which the baseline-LLM miscommunication rate increases significantly.

Being explicit about whether DECRYPTO is approached from a *generalist* or *specialist* perspective provides clarity and avoids unfair comparisons between models making inherently different assumptions. It also leads to two equally rich but distinct research directions. In the first case, DECRYPTO acts as a benchmark for multi-agent reasoning and measures progress in foundation models. In the second case, it is an RL environment used to research and evaluate training algorithms.

## 5  RESULTS

To kickstart this benchmark, we assess the cooperative, competitive, and theory of mind capabilities of specialist and generalist agents in Decrypto. Amongst generalist open-source models, we evaluate Llama-3.1-8B-Instruct, Llama-3.1-70B-Instruct (Dubey et al., 2024), and DeepSeek-R1-Distill-Qwen-32B (Guo et al., 2025). From the closed-source models, we evaluate GPT-4o, o1 and o3 (with reasoning set to "high"), as well as Claude 3.7 Sonnet with "Extended Thinking" for the human and ToM experiments. We also include the two baselines from Section 4 in all game playing experiments. We report mean ± standard error over three seeds. Additional details are in Section F.

**Baselines.**   We first look at the specialist word embedding baselines and show in Figure 2 that we can control $K$ for the top-$K$ selection to make them arbitrarily strong. Smaller $K$ prioritises words semantically similar to the keyword, making the hints easier to guess. Bigger $K$ selects

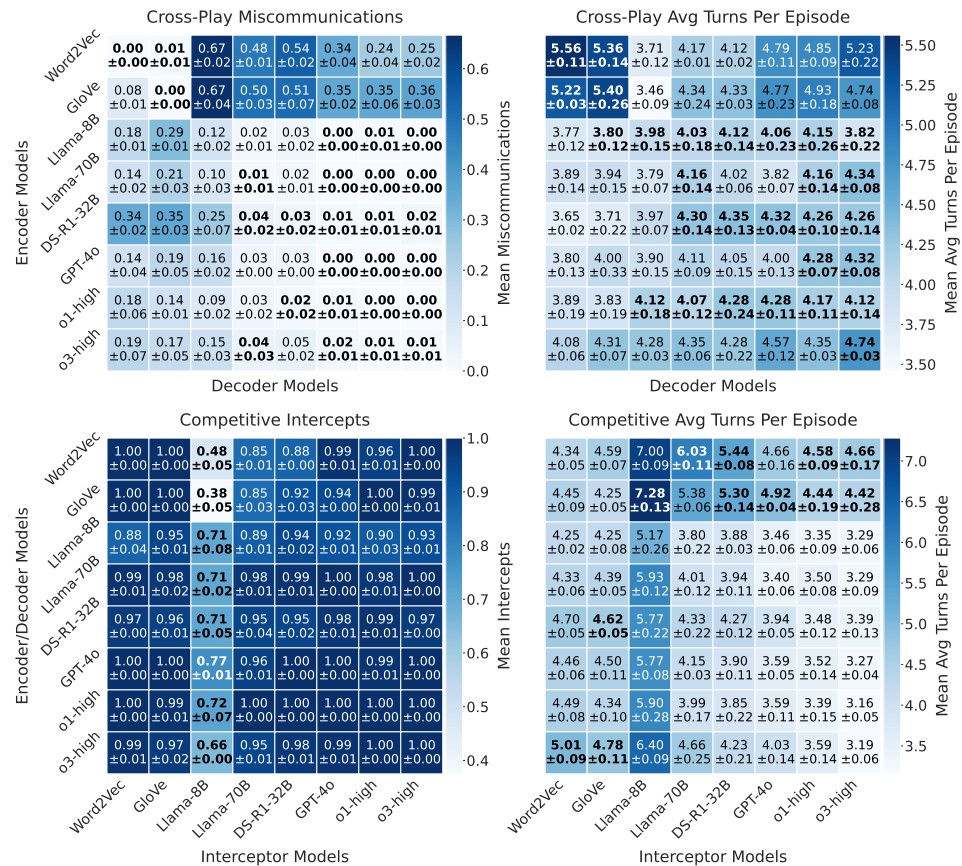

Figure 3: Cooperative and competitive results for 8 different agents. **Top:** Cooperative cross-play matrix when the interceptor is Llama 3.1-70B-Instruct. *Left:* proportion of games ending in miscommunication. *Right:* average number of turns per episode. LLMs coordinate well with each other, but lag behind baselines as encoders. **Bottom:** Competitive results playing a homogeneous encoder-decoder team against an interceptor. *Left:* proportion of games ending in an intercept. *Right:* average number of turns per episode. LLMs excel at intercepting other LLMs, but have short games and extremely low win rates as encoder-decoder, which we attribute to a failure of ToM.

semantically less similar hints, which are harder to guess if Bob cannot access the same embeddings. For example, suppose GloVe-Alice wants to hint the keyword "fire" to Word2Vec-Bob, and $K = 50$. The 50th most similar word to "fire" from the hint corpus would be "oil" for GloVe. However, "oil" is not even in the top 1000 for Word2Vec-Bob, highlighting how different word associations can produce cross-play difficulties. This finding also holds when an LLM plays Bob: as $K$ increases, miscommunications increase, and the average turn length decreases.

**Coordination.** The top row of Figure 3 shows the proportion of games ending in miscommunication and the average game length for different encoder-decoder pairs. Teams composed of an LLM and a baseline get significantly more miscommunications than baseline-only or LLM-only teams, evidence of different word associations between those two classes of models. LLM performance, measured by turns per episode, is positively correlated with model size or recency, but all LLM-only teams are outperformed by those where Alice is a baseline, highlighting the importance of hinting.

**Competition.** Looking at competitive game lengths in the bottom right of Figure 3, we see that LLMs are better than baselines at intercepting other LLMs, but are outmatched when intercepting baselines or when playing as the encoder-decoder team. The bottom left matrix shows that the vast majority of games end in interceptions, with win rates heavily skewed in Eve's favour. Even o3-high, the strongest LLM tested, achieves only $\sim 34\%$ win rate against the weakest interceptor, Llama-3.1 8B. While such a heavy bias may appear to be a property of the game, results in Section 5.1 suggest

| Model as Interceptor | Miscomms | Intercepts(↑) | Win Rate(↓) | Game Length(↓) |
|---|---|---|---|---|
| Word2Vec | $11.00 \pm 0.00$ | $8.00 \pm 0.00$ | $40.00\% \pm 0.00\%$ | $7.00 \pm 0.00$ |
| GloVe | $9.00 \pm 0.00$ | $9.00 \pm 0.00$ | $20.00\% \pm 0.00\%$ | $6.50 \pm 0.00$ |
| Llama 3.1-8B-Instruct | $11.67 \pm 0.33$ | $6.33 \pm 0.88$ | $46.67\% \pm 3.33\%$ | $6.97 \pm 0.13$ |
| ExploreToM-8B | $11.67 \pm 0.33$ | $4.00 \pm 2.31$ | $43.33\% \pm 6.67\%$ | $6.90 \pm 0.20$ |
| Llama 3.1-70B-Instruct | $11.33 \pm 0.33$ | $8.33 \pm 1.76$ | $46.67\% \pm 3.33\%$ | $6.80 \pm 0.21$ |
| DeepSeek-R1-Dist-32B | $9.33 \pm 0.33$ | $10.67 \pm 0.88$ | $30.00\% \pm 5.77\%$ | $6.33 \pm 0.03$ |
| GPT-4o | $10.67 \pm 0.88$ | $10.00 \pm 0.58$ | $36.67\% \pm 3.33\%$ | $6.50 \pm 0.15$ |
| o1-high | $9.33 \pm 0.33$ | $15.67 \pm 0.88$ | $20.00\% \pm 5.77\%$ | $6.23 \pm 0.12$ |
| o3-high | $6.67 \pm 0.33$ | $\mathbf{19.33 \pm 0.33}$ | $\mathbf{6.67\% \pm 3.33\%}$ | $\mathbf{5.50 \pm 0.15}$ |
| Claude 3.7 (+ ext. thinking) | $8.33 \pm 0.33$ | $14.00 \pm 0.58$ | $20.00\% \pm 0.00\%$ | $6.30 \pm 0.12$ |
| **Model as Decoder** | **Miscomms(↓)** | **Intercepts** | **Win Rate(↑)** | **Game Length(↑)** |
| Word2Vec | $20.00 \pm 0.00$ | $3.00 \pm 0.00$ | $0.00\% \pm 0.00\%$ | $2.70 \pm 0.00$ |
| GloVe | $20.00 \pm 0.00$ | $5.00 \pm 0.00$ | $0.00\% \pm 0.00\%$ | $3.10 \pm 0.00$ |
| Llama 3.1-8B-Instruct | $20.00 \pm 0.00$ | $4.33 \pm 0.88$ | $0.00\% \pm 0.00\%$ | $3.00 \pm 0.15$ |
| ExploreToM-8B | $20.00 \pm 0.00$ | $3.00 \pm 0.00$ | $0.00\% \pm 0.00\%$ | $2.53 \pm 0.09$ |
| Llama 3.1-70B-Instruct | $18.67 \pm 0.88$ | $7.67 \pm 0.33$ | $3.33\% \pm 3.33\%$ | $4.70 \pm 0.12$ |
| DeepSeek-R1-Dist-32B | $19.67 \pm 0.33$ | $6.67 \pm 1.20$ | $0.00\% \pm 0.00\%$ | $4.43 \pm 0.22$ |
| GPT-4o | $17.33 \pm 0.88$ | $8.67 \pm 0.67$ | $3.33\% \pm 3.33\%$ | $5.23 \pm 0.12$ |
| o1-high | $14.33 \pm 0.67$ | $8.00 \pm 1.00$ | $26.67\% \pm 3.33\%$ | $5.77 \pm 0.15$ |
| o3-high | $\mathbf{10.67 \pm 1.20}$ | $9.67 \pm 0.33$ | $20.00\% \pm 5.77\%$ | $6.30 \pm 0.12$ |
| Claude 3.7 (+ ext. thinking) | $12.67 \pm 0.67$ | $11.00 \pm 0.58$ | $16.67\% \pm 6.67\%$ | $6.57 \pm 0.12$ |
| Human (original games) | **11** | 12 | **40.00%** | **6.90** |

Table 1: We collect 10 games from human encoder-decoder teams and report the agents' performance when playing as (top) interceptor against human players or as (bottom) decoder with a human encoder. Reasoning models have a clear advantage, with o3-high standing out as a particularly strong interceptor. In ad-hoc human coordination, all agents perform worse than the human decoder, getting significantly more miscommunications, a lower win rate and shorter games. However, reasoning models again show stronger relative performance. ExploreToM-8B, a Llama 3.1-8B variant fine-tuned by Sclar et al. on ToM data, underperforms the latter in both settings, supporting the claim that DECRYPTO is complementary to other ToM benchmarks.

otherwise: that all LLMs tested make no attempt to consider Eve's perspective when hinting, instead choosing hints which will predictably get intercepted.

**Human Evaluation.** We collect 10 full games of human teams playing against a fixed LLM Eve, namely Llama 3.1-70B-Instruct. The players interacted with the game through a command-line interface and saw the same system prompts, user prompts and information as LLMs did. See Sections I and J for prompts and Section K for data collection details and the code repository for the data itself.

Table 1 shows the result of replaying collected human games by re-using the human-provided hints and replacing either Eve or Bob with different LLMs. Closed-source reasoning take the lead in both roles, with o3-high standing out as a particularly strong interceptor. As Bob, recent models perform best, coming close but not yet matching human decoders. The o-series models were also particularly verbose: it is the only experiment in which they occasionally exceeded 10k tokens per response. We also evaluate ExploreToM-8B (Sclar et al., 2024), a variant of Llama 3.1-8B fine-tuned on ToM tasks, which significantly outperforms the latter on multiple ToM benchmarks. Despite that, we found it to perform slightly worse in our DECRYPTO experiments, which **demonstrates that our benchmark is complementary to existing Sally-Anne type tasks.**

## 5.1 THEORY OF MIND

Drawing upon seminal works in cognitive psychology, we adapt classic ToM experiments to expose the limitations of LLMs and explain their poor performance in DECRYPTO.

The first experiment adapts the Smarties Task of Gopnik & Astington, which presents children with a deceptive object (a box of Smarties containing pencils) and studies whether the child can correctly identify incorrect beliefs – either their own or of another child – when first encountering the object. To recreate this task in DECRYPTO, we substitute the closed Smarties box and the pencils with the game history and the secret keywords. At each turn except the first, we prompt Eve three times independently. Prompt $A$ asks her to predict the four keywords. Prompt $B$ reveals the keywords and

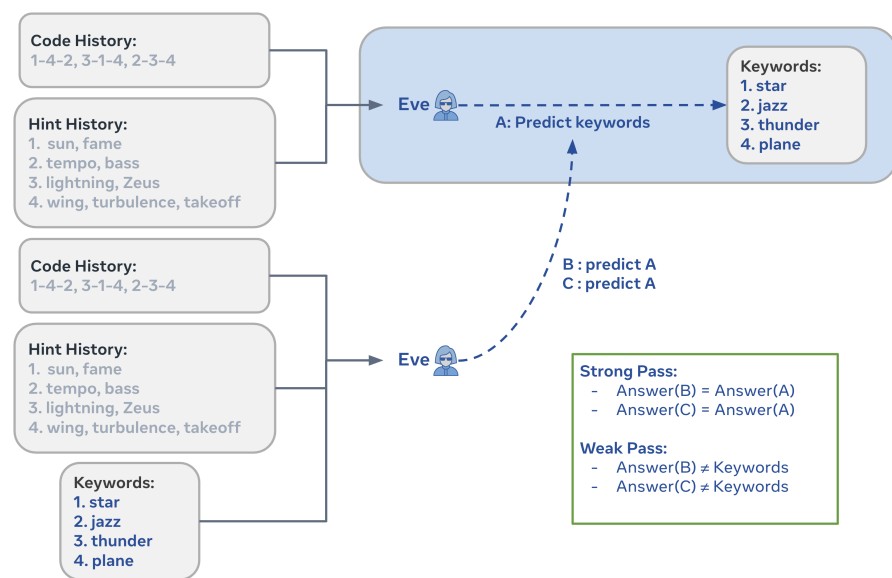

Figure 4: Diagram of the ToM experiment evaluating Representational Change and False Belief. Eve is prompted three times. A) She is prompted to predict the keywords given the code and hint histories. B) Eve is revealed the keywords and is asked what she thought were the four keywords pre-reveal. C) Eve is also revealed the keywords and is asked what a second interceptor would think are the keywords pre-reveal. For B or C, we consider it a strong pass if the answer matches the answer to A. We consider it a weak pass if he answer is *not* the keywords. Because we only evaluate on turns where Answer(A) $\neq$ Keywords, a strong pass always implies a weak pass as well.

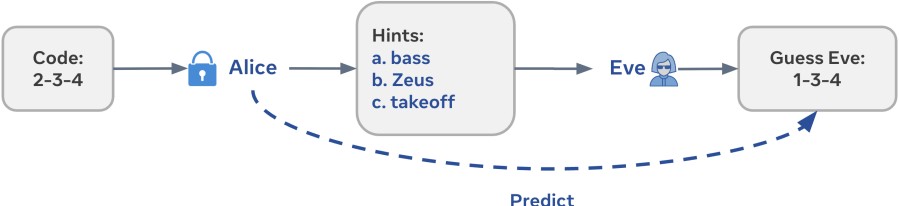

Figure 5: Diagram of the ToM experiment evaluating perspective taking. After Alice has given her hints, we prompt her to predict Eve's guess. This is done in branching context, and therefore does not change Alice's behaviour in the game.

asks Eve what she thought were the keywords pre-reveal. Prompt $C$ also reveals the keywords but asks the model to predict what a "second interceptor" would think the keywords to be, pre-reveal.

We generate outputs with temperature 0 (for models that allow it) and only consider turns where the answer to $A$ is an incorrect guess, keeping only cases where Eve has inaccurate "beliefs" pre-reveal. We compare answers $A$ and $B$ to measure **representational change** (RC), the ability of the agent to recognise when its belief about the world (but not the world itself) changes due to additional information. Similarly, comparing $A$ and $C$ measures **false belief** (FB), the ability to represent other agents as having false beliefs about the world. We distinguish two variants of the tasks. The *Weak* variant only requires the agents to realise that either themselves or the second interceptor could not have known the ground truth, and so an answer to either $B$ or $C$ is correct as long as it differs from the real keywords. For the *Strong* variant of those tasks, we consider the agent to pass only if it correctly predicts its answer to prompt $A$ (i.e. if $B = A$ or $C = A$). Success here likely requires a

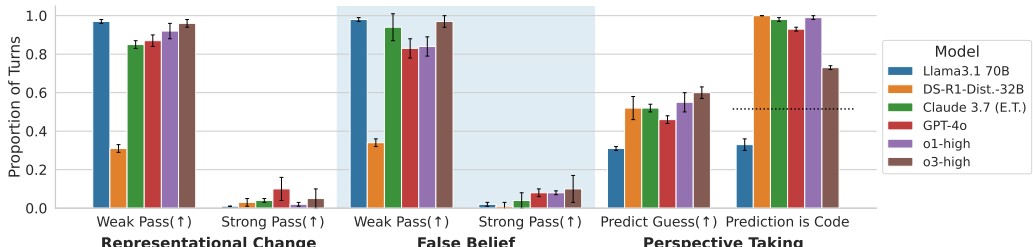

Figure 6: *Representational change* (RC), *false belief* (FB) and *perspective taking* (PT) scores in DECRYPTO. RC measures whether agents can identify a change in their own world-view following new information. FB measures whether agents can model the incorrect beliefs of another agent. PT assesses whether LLMs can reason from the point of view of an agent with less information. Most models perform well on the weak RC and FB tasks, but fail on the strong variants, demonstrating a lack of self-consistency or counterfactual reasoning. In PT, all models except Llama 3.1-70B and o3-high erroneously predict that Eve will intercept on nearly every turn, including the first. Models also fail to consider Eve's point of view by choosing hints they subsequently predict will get intercepted.

self-consistent representation of the keywords, or at least strong counterfactual reasoning. Figure 4 summarises the procedure.

The first two thirds of Figure 6 evaluates the RC and FB abilities of six different LLMs. Most models tested achieve high accuracy in both weak RC and weak FB, with Llama 3.1-70B slightly outperforming newer models, including reasoning ones. That said, models struggle to correctly identify which belief they or another agent held prior to receiving new information, as evidenced by the $\leq 10\%$ accuracy on the strong variants.

The second experiment measures **perspective taking** (PT) and is a word-based code-guessing variant of the Three Mountain Problem (Piaget et al., 1956). After Alice has given her hints, we prompt her to predict Eve's guess, as shown in Figure 5. We record the prediction accuracy and the proportion of turns on which Alice predicts that Eve will guess the correct code. This last metric is particularly insightful as it explains why LLMs struggle to avoid getting intercepted in Section 5. On the one side, if Alice predicts that Eve will intercept more rarely than she does means **Alice underestimates Eve's abilities or fails to account for all the information available to her**, which in turn can lead Alice to give hints that are too obvious. This is the case for Llama-3.1 70B, on the right of Figure 6. It trails behind other models in prediction accuracy, and is also the only model to predict fewer interceptions than ground truth (the real interception rate is $\sim 52\%$, indicated by the dotted line).

On the other side, if Alice predicts that Eve will intercept on nearly every turn, it is a **failure to reason from Eve's perspective**. It would indicate that LLMs confuse their knowledge with that of others, just like young children in Piaget et al.'s study, by either explicitly or implicitly assuming that Eve has access to the keywords. It also shows a **second failure: that of integrating ToM reasoning in decision-making**. Indeed, if Alice thinks Eve will guess the code given her hints, she should have chosen different hints in the first place.

Such a failure case is precisely what we observe with DeepSeek-R1-Distill-Qwen-32B, Claude 3.7 Sonnet, GPT-4o and o1-high. All those models predict that Eve will intercept on nearly every turn, *including the first,* as evidenced by model outputs in Section M. Only Llama and o3-high correctly point out that Eve can do no better than a random guess, albeit o3-high still overestimates the rate of interceptions. Surprisingly, Listing 16 shows these results to hold even if we modify the PT prompt to emphasize that Eve "*does *NOT* know the secret keywords*", with o1-high showing no statistical difference in predictions with the modified vs. the original prompt. This provides a diagnostic for why interceptions in DECRYPTO are so common: **LLMs do not attempt to model other agents before making decisions, and also struggle to model them if explicitly asked to.**

## 6 RELATED WORK

BattleAgentBench (Wang et al., 2024), multi-robot studies (Li et al., 2023) and other works such as Liu et al. (2023) and Guo et al. (2024) have recently studied LLMs in multi-agent contexts. While detailed, they focus on spatial reasoning rather than on the language-based reasoning and ToM abilities central to DECRYPTO. Among multi-agent environments, social deduction games

such as Werewolf (Xu et al., 2023b;c), Avalon (Light et al., 2023), Chameleon and Undercover (Xu et al., 2024b) are particularly amenable to studying LLMs, as are negotiation games (Abdelnabi et al., 2023). However, they provide less structured and atomic insight than DECRYPTO, given the free-form dialogue and frequent deception in those settings. AgentPro (Zhang et al., 2024b) studies competitive scenarios in Blackjack and Poker, but lacks the cooperative aspect of DECRYPTO.

The study of artificial ToM started with ToM-net (Rabinowitz et al., 2018), a simple network capable of predicting the behaviour of other artificial agents and acting based on false beliefs. In contrast, recent works on ToM have focused on LLMs modelling human beliefs (Chen et al., 2024; Sclar et al., 2024; Kim et al., 2023). DECRYPTO connects both by enabling the study of LLM ToM applied to both humans and other artificial agents. Unlike other recent ToM benchmarks (Le et al., 2019; He et al., 2023; Kim et al., 2023; Zhou et al., 2023; Xu et al., 2024a; Gandhi et al., 2024; Sclar et al., 2024) and Rabinowitz et al. (2018), however, DECRYPTO does so in an interactive setting, where agents can not only model the beliefs of other agents, but also influence them.

Kosinski (2023) also studied the Smarties Task, from which they concluded that "ToM-like ability [...] may have spontaneously emerged" in LLMs. Those claims were directly challenged by Ullman in subsequent work, with which our results agree. Shaikh et al. (2023) use another language game, Codenames, to study cross-cultural pragmatics. Their work stops short of evaluating LLMs, but raises important questions about the role of cultural backgrounds in pragmatic inference. DECRYPTO is a natural choice for pursuing research in this direction, since we can specify different personas in the LLM prompts.

## 7 CONCLUSION

Our paper introduces DECRYPTO, an interactive multi-agent benchmark designed to evaluate coordination, competition and theory of mind capabilities in language models. DECRYPTO addresses critical limitations of existing ToM benchmarks, such as biases arising from textual translation of embodied scenarios or lack of interactivity. Designed to be future-proof and to eliminate confounding factors known to limit LLM performance, DECRYPTO fills an important gap in existing benchmarks. Furthermore, our codebase provides a versatile platform for quickly designing interactive ToM experiments inspired by cognitive psychology.

We conduct extensive experiments to evaluate open-source and closed-source LLMs. We find that even state-of-the-art models struggle with the nuanced communication and strategic reasoning that DECRYPTO requires, often underperforming simple baselines in cooperative and competitive settings alike. Similarly, our human-AI experiments shed light on the limited ability of recent LLMs to coordinate with humans or understand their communications.

Finally, we create variants of two seminal experiments from cognitive psychology to study LLM ToM in interactive settings, analogously to how it is studied in children. Our experiments provide strong evidence that state-of-the-art models still lack many ToM skills. In particular, we identify their inability to reason from the perspective of other agents as the main cause behind their poor performance in DECRYPTO and predict that this same inability will also limit their deployment in other multi-agent scenarios with partial information.

DECRYPTO opens up several promising research directions. On the ToM side, these include enhancing the ToM capabilities of language models and investigating differences between how LLMs model humans versus how they model other agents. From the multi-agent perspective, an exciting avenue for future work is adapting algorithms from multi-agent RL (Silver et al., 2017; Hu et al., 2020) to fine-tune LLMs. For a longer discussion on the limitations of our work and on the multiple research directions enabled by DECRYPTO, we refer the reader to Section A.

## REPRODUCIBILITY STATEMENT

We have taken several steps to enable reproduction of our results. Our source code and collected human data are available at `anonymous.4open.science/r/decrypto/`. This includes detailed instructions and configuration files to reproduce the main results presented in our paper.

Experimental details and prompts are provided in Sections I to K. Detailed derivations of our theoretical results linking DECRYPTO to pragmatic inference are in Section H.

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

## A    LIMITATIONS AND FUTURE WORK

We firmly believe DECRYPTO to be a valuable and future-proof benchmark for evaluating multi-agent reasoning and theory of mind. However, our work still has a number of limitations, which we hope to address in future work.

First, while we largely eliminated the confounding factors that affect LLM performance, the game dynamics and the ToM tasks may still be too complicated for weaker models. For instance, when trying Llama 3.1-8B-Instruct on the perspective taking task, we found that it failed to understand the prompt altogether.

Regarding evaluations, the main limitation of our work is the computational cost of computing Figure 3. An individual run is relatively cheap (approximately $0.10 per role per episode for Claude 3.7 Sonnet). However, given the 3-player nature of the game, the total number of combinations for $n$ models is $n^3$, or $2n^2$ if only computing the matrix slices in Figure 3. For a large number of models, this can induce significantly larger costs than standard single-agent benchmarks. To mitigate this, we provide the tools to run evaluations locally and efficiently, leveraging prefix caching where possible to reduce unnecessary computations. We also acknowledge the limited scope of our human study in Table 1. While the 10 games collected were sufficient to demonstrate the limitations of LLMs in human-AI interactions, we believe that there is much to be learned from a more extensive study, including with humans as interceptors. We therefore made sure to provide the tools to enable such research, such as a lightweight command-line interface for conducting human studies.

Finally, theory of mind is an umbrella term for multiple abilities. Our work evaluates some of them, such as representational change, false belief and perspective taking, and we hope extensions to DECRYPTO to expand on this set in the future. Nevertheless, there are ToM abilities which DECRYPTO is ill-fitted to evaluate. For instance, models performing well in the game and the ToM tasks we designed may still fail to reason about the intentions or emotions of other agents.

DECRYPTO also enables a number of other research directions connected to ToM, multi-agent reasoning and reinforcement learning, and pragmatic inference. From a ToM perspective, DECRYPTO is easy to extend with additional ToM experiments, which can complement the results presented in our work. Given the limitations of current frontier models, we also believe there is a dire need for methods to improve the ToM abilities of LLMs. DECRYPTO features short episodes and dense rewards, making it a suitable environment to study the application of multi-agent RL algorithms towards LLM post-training. In particular, DECRYPTO features stochasticity and partial observability, two key ingredients for "meaningful" coordination challenges (Ellis et al., 2023). It would therefore be interesting to study whether fine-tuning LLMs on DECRYPTO using self-play (Silver et al., 2017) or other MARL algorithms (Hu et al., 2020) produces the same kind of idiosyncrasies observed when training *tabula rasa* models in other cooperative environments (Cui et al., 2023; Gessler et al., 2025). Finally, we believe the study of pragmatic inference in LLMs to be a promising avenue for future research, which is made much easier by the release of our benchmark. We formalise DECRYPTO as a pragmatic inference game and provide suggestions for this line of work in Section H.

## B    IMPACT STATEMENT

As LLMs become more prevalent in our daily lives and start interacting with each other and with human users, we believe it becomes crucial to understand the multi-agent and ToM abilities of foundation models. Our work provides a novel assessment of those abilities, and the tools to conduct future studies on inter-AI and human-AI interactions. As such, we hope to bring attention to what we consider to be an understudied subset of agentic capabilities, and that our work will lead to models with a strong social and human-centric behaviour.

However, ToM encompasses a broad set of abilities, of which DECRYPTO evaluates only a subset. A possible misuse of DECRYPTO is to equate strong performance on our benchmark to more general ToM abilities, including intent inference and emotion understanding, which are not evaluated by DECRYPTO. Such claims have incorrectly been made about other ToM benchmarks (Hu et al., 2025), and so we ask future researchers to be careful when concluding that an AI system has ToM.

## C    RATIONALE FOR THE CODE IN FIGURE 1

The code that Alice encrypted in Figure 1 is `[4-1-3]`. The rationale is at follows. The four keywords are `[1.  star, 2.  jazz, 3.  thunder, 4.  plane]`, and the hints are `[two, clone, AC/DC]`. "Two" refers to "two dimensions", which is the defining characteristic of a geometric plane. "Clone" is a hint for "star", since both "clone" and "star" are common operations performed on a GitHub repository. "AC/DC" is a rock band, and one of their most famous songs, "Thunderstruck" starts with the word "Thunder" being chanted repeatedly.

The choice of hints relies on multiple assumptions made by Alice, here played by the authors, about Bob, played by the readers. The first hint relies on the assumption that many readers will have a mathematical background and quickly think of "plane" in its geometric sense, rather than as an aircraft. Similarly, we assume for the second hint that readers will be sufficiently familiar with GitHub to make the same connection as us. Failing that, we also anticipated that some readers might link the two terms by thinking of the "clone troopers" from Star Wars, which also leads to a correct guess. Finally, we assumed that the song "Thunderstruck" is famous enough to be picked up by most readers.

This is an example of how Alice can use ToM and the specific information she has about another player, in this case the readers, to inform her choice of hints. We hope we were successful.

## D    BASELINE DETAILS

Here we provide additional details on the word-embedding baselines introduced in Section 4.

The baselines work as follows. Alice can choose hints from a fixed corpus of the 5696 most common nouns extracted from the Brown, Gutenberg, and Webtext corpora (Bird et al., 2009), filter to ensure all extracted nouns are present in both the GloVe and Word2Vec embeddings. (We use this hint corpus for the baselines only; LLM agents generate hints in an open-ended fashion.)

Before Alice chooses hints, Alice and Bob agree on a strategy. For each code digit $c_i \in [1, 4], i \in [1, 3]$, Alice picks a hint $h_i$ and guarantees that the hint picked for the digit is more similar to the $c_i$-th keyword $k_{c_i}$ than to any other keyword, i.e., $s(h_i, k_{c_i}) > s(h_i, k_j) \quad \forall j \in -\mathbf{c}$, where $-\mathbf{c} = [1, 4] \backslash c_i$ and $s$ is the cosine similarity.

Alice increases the diversity of her hints by randomly sampling each hint from the top-$K$ most similar words for each keyword under cosine similarity, filtering those that do not satisfy the constraint above. We also enforce that Alice cannot reuse any previously used hints in the same episode.

Bob then analyses the hints and assigns each hint to its most similar keyword under cosine similarity, i.e., $\arg\max_{c \in [1,4]} s(h_i, k_c) \ \forall i \in [1, 3]$. Bob guesses the code perfectly if Alice and Bob use the same word embeddings because the hints are guaranteed to be the most similar to the correct keyword under the same embedding model.

Eve is only given the hints $\mathbf{h}$ and the hint history for each keyword, which at turn 0 is empty. Eve calculates the cosine similarity between the average embedding of each keyword's hint history with the hints. This results in a similarity matrix of size $N \times M$, where $N(= 3)$ is the number of hints and $M(= 4)$ is the number of keywords. Eve combinatorially calculates the globally optimal guess based on the similarity values. Since $N$ and $M$ are typically small values, we are unconcerned about the computational complexity. However, Eve's selection task is equivalent to a linear assignment problem and efficient algorithms, such as the Jonker-Volgenant algorithm, are available in open-source libraries such as SciPy (Virtanen et al., 2020).

## E  DIAGRAMS FOR THEORY OF MIND EXPERIMENTS

[Diagrams moved to the main text. Keeping Appendix header for now to not change the indexing of other Appendices. It will be removed for the camera-ready copy.]

## F  EXPERIMENTAL DETAILS

Here we provide details for the experiments reported in Section 5.

### F.1  PROMPTING

All our experiments are text-only and conducted exclusively through prompts, which we provide in Sections I and J. The system prompt is composed of an explanation of the game rules, common to all roles, and a set of role-specific instructions. The user prompts are role-specific but follow a similar structure, providing, in order, a summary of the previous turn and of the game so far, role specific information (e.g. the code or the hints), a summary of the role task, and formatting instructions. For the ToM tasks, the prompts are structure similarly, and rely on the same output formatting. For agents that do not admit a system prompt, such as Deepseek-R1, we include the system prompt as a prefix to the first user message.

We maintain prompts and model outputs in context, making DECRYPTO a multi-turn environment. We use a regular expression to extract the hints or the guess from the model outputs. If the code fails to detect a correctly-formatted answer in the output, we re-prompt the model again up to 10 times, reminding it of the correct output format. If the model has still not provided an answer after 10 attempts, we default to a dummy answer. In practice, this almost never happens unless the output token limit is set very low for very verbose models, such as Deepseek-R1. By default, retries are not added to the model's context to avoid cluttering it and unnecessarily lowering its performance. ToM prompts and outputs are also excluded from the context for future turns.

### F.2  GENERATION PARAMETERS

We set the model temperature to 0.6 for experiments in Figure 3 and table 1, and to 0 for ToM experiments in Figure 6 to reduce the impact of model stochasticity. We note that some close source models are not entirely deterministic, even with the temperature set to 0.

Our goal was not to evaluate model verbosity, but rather the limit of their capabilities. Therefore, we set generous token limits (between 750 for non-reasoning models and up to 10000 for reasoning ones) to prevent cutting model generations prematurely. We found that models varied widely in their verbosity. For instance, Claude 3.7 Sonnet with Extended Thinking output approximately 500 total tokens per turn on average, and sometimes as low as 350 despite a limit of 1500 tokens. In contrast, some responses by o1-high and o3-high exceeded 8000 tokens.

The full set of generation parameters can be found in the configs we provide as part of our code release, and which enable reproduction of the results in our paper with only a few commands.

### F.3  COMPUTATIONAL COSTS

We ran our experiments on a mix of local and API-hosted models.

All open-source models were hosted locally on a high performance computing cluster using vLLM (Kwon et al., 2023). Each model instance ran on 8x Nvidia A100 GPUs. Although not necessary, we also launched multiple instances of the same model running, using concurrency to accelerate inference.

System prompts are on the order of 1000 tokens, and the user prompts at each turn are less than 300 tokens each. Models will typically output around 500 tokens per prompt, but the number can go much higher for particularly verbose models. We estimate that evaluating Claude 3.7 Sonnet with Extended Thinking cost approximately $0.10 per role per episode. A full run of 32 episodes, 3 random seeds, and with all three roles played by Claude 3.7 Sonnet costs approximately $30.

Finally, we detail human data collection in Section K.

## G    ADDITIONAL RESULTS

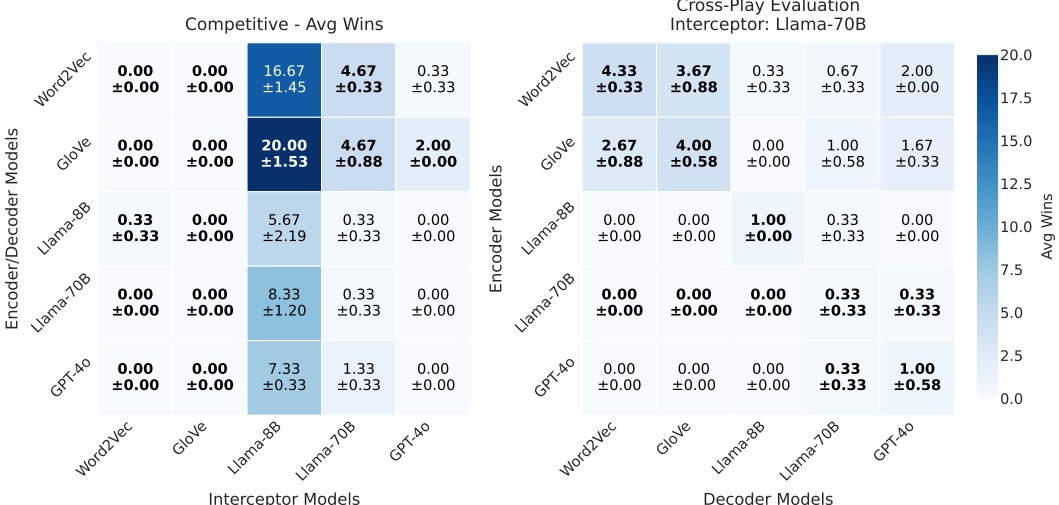

Figure 7: Number of survived games over 32 games for 5 different agents. Mean and standard deviation computed over 3 model seeds. We find the abilities of the current generation of LLMs to be skewed, such that Eve wins a vast majority of the games. As models improve and can come up with better hints, we expect the balance to shift.

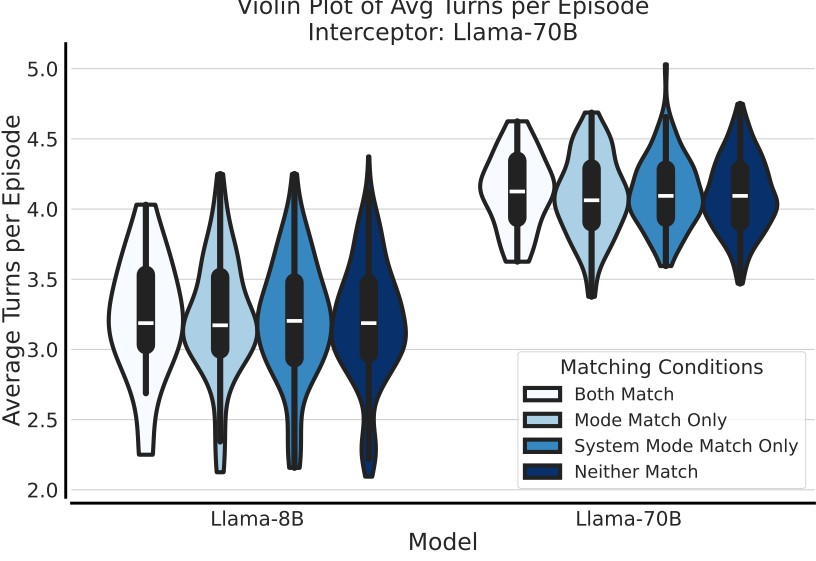

Figure 8: Distribution of the number of turns per game as we vary the system and user prompts for both encoder and decoder. We handwrite 5 system and 5 user prompts (i.e. "modes") for the encoder and the decoder and plot the distributions of games when only their system prompts match, when only their user prompts match, when both match or when neither do (625 total combinations per model). We find that both models tested are robust to significant prompt variations, with model size remaining the most significant predictor of performance.

**Robustness.**    We show that prompt variants do not significantly affect the final performance measured by average turn length for Llama-3.1-8B and Llama-3.1-70B, see Figure 8. This suggests that

the poor performance of LLMs in DECRYPTO is more likely due to a lack of reasoning abilities than a lack of prompt tuning. We look at the robustness of different models to variations in prompts and generation parameters, keeping in mind that generalist agents cannot control their prompt since it is assumed to be part of the environment. We handcraft 5 system and user prompts for Alice and Bob, respectively. The system prompt consists of 2 components. One component is responsible for explaining the game rules in general, for which we have 5 variants. The second component explains the specific role. We have 5 prompt variants for each role, already resulting in 125 different prompt combinations. The user prompt instructs the specific roles to take their actions, for which we have 5 variants. In total, this results in 625 different prompt setups for each model. We run 32 games over 3 model seeds per system/user prompt combination and measure the average game length.

## H    DECRYPTO AS A PRAGMATIC INFERENCE GAME

With mild assumptions, we can formalize DECRYPTO as an iterated pragmatic inference game in the Rational Speech Act (RSA) framework introduced by Frank & Goodman and extended in Goodman & Frank (2016). Doing so provides a Bayesian view of how successful communication in the game requires agents to model each other's decision process.

We consider each turn in isolation, so that each episode of DECRYPTO is composed of up to 8 independent pragmatic inference games. Agents must associate utterances $u$ (i.e. a set of 3 hints) to an intended meaning $m$ (the code for that turn or, equivalently, the corresponding combination of 3 keywords).

The RSA framework starts by defining a hypothetical *literal listener*, which interprets utterances according to their semantics:

$$P_{\text{Lit}}(m|u) \propto \delta_{m \in [[u]]} P(m). \tag{1}$$

Here $P(m)$ is the probability of a given meaning and $\delta_{m \in [[u]]}$ is the publicly known *lexical function*, which returns 1 if the meaning $m$ is *semantically compatible* with utterance $u$ and 0 otherwise. The literal listener therefore assigns 0 probability to meanings that are semantically incompatible with the utterance, but otherwise defers to the normalized prior probability of a meaning. In DECRYPTO, all meanings are equally likely, so $P(m) = \frac{1}{|M|}$, where $M$ is the set codes that have not been seen on previous turns.

The game rules stipulate that hints must refer to the meaning of the keywords. Therefore, $[[u]]$ can be defined as the subset of all combinations of three keywords $m \in M$ that have a semantic link to the hints. For instance, suppose the keywords are [star, jazz, thunder, plane], as in Figure 1, and the hints are $u' = $ [fusion, Zeus, pilot]. Then,

$$m_1 := [\text{star, thunder, plane}] \equiv [1, 3, 4] \in [[u']], \tag{2}$$

$$m_2 := [\text{jazz, thunder, plane}] \equiv [2, 3, 4] \in [[u']], \tag{3}$$

among other meanings, and $P_{\text{Lit}}(m_1|u') = P_{\text{Lit}}(m_2|u') = \frac{1}{|[[u']]|}$. This is because stars are fuelled by nuclear fusion and "jazz fusion" is a popular music genre.

Alice is then assumed to be an *approximately rational speaker*, who chooses utterances according to a probability

$$P_{\text{Alice}}(u|m) \propto e^{\lambda U(u,m)}, \tag{4}$$

where $\lambda$ captures the degree of optimality and $U(u, m)$ is the utility of speaking utterance $u$ to communicate meaning $m$. To avoid infinite recursion, the speaker's *utility* in RSA balances the utterance cost with the probability that the literal listener will interpret the utterance correctly:

$$U(u, m) = \log P_{\text{Lit}}(m|u) - \text{cost}(u, m).$$

The RSA framework typically assumes a single speaker and collaborative listener, but we can capture the competitive dynamics between Alice and Eve by letting the utterance cost be a function of the probability that Eve intercepts the code. There are multiple possible candidate functions, but since miscommunication and intercepts play an equally important role in DECRYPTO, we let

$$\text{cost}(u, m) = \varepsilon \log(1 - P_{\text{Eve}}(m|u)). \tag{5}$$

Here, $P_{\text{Eve}}(m|u)$ denotes Eve's probability of intercepting the code and $\varepsilon \in [0, 1]$ captures how bad an intercept would be on that turn. For instance, if this is the eighth turn and no intercept has occurred so far in the game, getting intercepted will not impact Alice and Bob's chances of winning, so we set $\varepsilon = 0$. Similarly, we can add a variable $\beta \in [0, 1]$, which captures the importance of not miscommunicating on this turn. This makes Alice's utility

$$U(u, m) = \beta \log P_{\text{Lit}}(m|u) + \varepsilon \log(1 - P_{\text{Eve}}(m|u)). \tag{6}$$

Finally, Bob, the *pragmatic listener*, decodes utterances based on the common prior over meanings and to a model of the speaker, with probability

$$P_{\text{Bob}}(m|u, \text{Alice}) \propto P(m)P_{\text{Alice}}(u|m) \tag{7}$$

Expanding Equation (7), we get

$$P_{\text{Bob}}(m|u, \text{Alice}) \propto \frac{1}{|M|}e^{\lambda U(u,m)} \tag{8}$$

$$= \frac{1}{|M|}e^{\lambda(\beta \log P_{\text{Lit}}(m|u) + \varepsilon \log(1 - P_{\text{Eve}}(m|u)))} \tag{9}$$

$$= \frac{1}{|M|}P_{\text{Lit}}(m|u)^{\lambda\beta}(1 - P_{\text{Eve}}(m|u))^{\lambda\varepsilon}. \tag{10}$$

Hence, in all cases where $\varepsilon > 0$, **Bob must model how Alice models Eve in order to interpret the hints correctly**. This is a clear example of why multi-step theory of mind is necessary for optimal play in DECRYPTO.

The listener rarely has a perfect model of the speaker. Goodman & Frank discuss *uncertain RSA* (uRSA), a common extension to the RSA framework where the pragmatic listener is uncertain about the model underlying the speaker's behaviour (Kao et al., 2014). Under uRSA, the pragmatic listener has a distribution over possible speakers $s$, and so Bob has a joint probability over speaker and interpretation:

$$P_{\text{Bob}}(m, s|u) \propto P(m)P(s)P_s(u|m, s). \tag{11}$$

Marginalizing over speakers gives us the probability that Bob interprets the hints correctly:

$$P_{\text{Bob}}(m|u) \propto P(m)\sum_s P(s)P_s(u|m, s) = P(m)P_{\text{marginal}}(u|m), \tag{12}$$

with $P_{\text{marginal}}(u|m) := \sum_s P(s)P_s(u|m, s)$. Comparing this to equation 7 highlights the fact that Bob's probability of interpreting the hints correctly depend on how close the marginal probability $P_{\text{marginal}}(u|m)$ is to the real utterance probability $P_{\text{Alice}}(u|m)$. This provides an entry point for studying cross-cultural pragmatics in DECRYPTO, expanding upon work done by Shaikh et al. (2023) in Codenames: if Alice provides hints informed by her cultural background, but such background is unlikely under Bob's distribution, we expect the number of miscommunications between them to increase significantly.

## H.1 MODELLING EVE

Under the RSA framework, Alice's degree of optimality is determined by $\lambda$, which can be thought to capture the bounded rationality of the speaker. However, our definition of cost in equation 5 introduces a second failure mode in the ability to model Eve accurately.

To show this, suppose Alice only has access to a proxy model $\tilde{P}(m|u)$ of Eve, rather than to her true probabilities. Alice therefore chooses utterances according to probability $\tilde{P}_{\text{Alice}}(u, m)$, which follows the same form as equation 4, except using the proxy utility

$$\tilde{U}(u, m) = \beta \log P_{\text{Lit}}(m|u) + \varepsilon \log(1 - \tilde{P}(m|u)). \tag{13}$$

We can use this to derive Alice's expected (real) utility for each meaning $m$:

$$\mathbb{E}\{U(m)\} = \sum_{u \in \mathcal{U}} \tilde{P}_{\text{Alice}}(u|m)U(u, m) \tag{14}$$

$$= \sum_{u \in \mathcal{U}} \frac{P_{\text{Lit}}(m|u)^{\lambda\beta}(1 - \tilde{P}(m|u))^{\lambda\varepsilon}}{Z(m)}[\beta \log P_{\text{Lit}}(m|u) + \varepsilon \log(1 - P_{\text{Eve}}(m|u))], \tag{15}$$

where $Z(m) := \sum_{u'} P_{\text{Lit}}(m|u')^{\lambda\beta}(1 - \tilde{P}(m|u'))^{\lambda\varepsilon}$.

For simplicity, we let $\varepsilon = 1$ and we can eliminate the terms depending on $P_{\text{Lit}}(m|u)$. This corresponds to scenarios where $\beta = 0$ because miscommunication does not matter, or where $P_{\text{Lit}}(m|u) = 1$ for all $u$ under consideration. The latter is not as strong an assumption as it seems. The space of all possible utterances, even if restraining hints to be single words, is equal to the entire English vocabulary cubed. Constraining Alice to utterances with a unique interpretation given the keywords (s.t. $|[[u]]| = 1$) is therefore reasonable, and still allows for a very large number of possible utterances. In fact, this is very similar to the way the word embedding baselines in Figure 2 only choose hints that are guaranteed to be interpreted correctly by Bob. Given these conditions, the task becomes exclusively to minimize the probability of interception:

$$\mathbb{E}\{U(m)\} = \sum_{u\in\mathcal{U}} \tilde{P}_{\text{Alice}}(u|m) log(1 - P_{\text{Eve}}(m|u)) \tag{16}$$

$$= \sum_{u\in\mathcal{U}} \frac{(1 - \tilde{P}(m|u))^{\lambda}}{Z(m)} \log(1 - P_{\text{Eve}}(m|u)). \tag{17}$$

Under the same simplifications, the utterance probability under the RSA framework if Alice had access to the real $P_{\text{Eve}}(m|u)$ is given by

$$P_{\text{true}}(u|m) := \frac{(1 - P_{\text{Eve}}(m|u))^{\lambda}}{Z_{\text{true}}(m)}, \tag{18}$$

where $Z_{\text{true}}(m) := \sum_{u'}(1 - P_{\text{Eve}}(m|u'))^{\lambda}$.

Combining equation 16 and equation 18, we obtain

$$\mathbb{E}\{U(m)\} = \sum_{u\in\mathcal{U}} \tilde{P}_{\text{Alice}}(u|m)\frac{1}{\lambda}[\log(P_{\text{true}}(u|m)) + \log(Z_{\text{true}}(m))] \tag{19}$$

$$= \frac{1}{\lambda} \sum_{u\in\mathcal{U}} \tilde{P}_{\text{Alice}}(u|m)[\log(\frac{P_{\text{true}}(u|m)}{\tilde{P}_{\text{Alice}}(u|m)}) + \log(\tilde{P}_{\text{Alice}}(u|m)) + \log(Z_{\text{true}}(m))] \tag{20}$$

$$= -\frac{1}{\lambda} D_{\text{KL}}(\tilde{P}_{\text{Alice}}||P_{\text{true}}) - \frac{1}{\lambda} H(\tilde{P}_{\text{Alice}}) + \frac{1}{\lambda} Z_{\text{true}}(m), \tag{21}$$

with $D_{\text{KL}}(\cdot||\cdot)$ denoting the Kullback–Leibler (KL) divergence and $H(\cdot)$ the entropy.

Therefore, for a given $\lambda$, the log likelihood of getting intercepted by Eve depends linearly on the KL divergence between $\tilde{P}_{\text{Alice}}$ and $P_{\text{true}}$, where

$$\tilde{P}_{\text{Alice}}(u|m) \propto (1 - \tilde{P}(m|u))^{\lambda}$$

$$P_{\text{true}}(u|m) \propto (1 - P_{\text{Eve}}(m|u))^{\lambda}.$$

From this it follows that expected utility is only maximised if $\tilde{P}(m|u) = P_{\text{Eve}}(m|u) \,\forall u$. In the limit of $\lambda \to \infty$, Alice concentrates her utterance probability on $u^* := \arg\max_u \tilde{U}(u,m)$. In such case, the expected utility is given by

$$\lim_{\lambda\to\infty} \mathbb{E}\{U(m)\} = \log(1 - P_{\text{Eve}}(m|u^*)). \tag{22}$$

This demonstrates that even if miscommunication is no issue and Alice is perfectly rational, she still incurs a risk of interception and a lower utility if her model of Eve is inaccurate.

However, we note that an imperfect model of Eve does not fully account for the behaviour we observe in LLMs. Indeed, our ToM results in section Section 5 show that LLMs not only fail to predict Eve's guess, they often fail to consider the cost of intercepts altogether when choosing hints.

### H.2 FURTHER REFINEMENTS

Our formalism assumes $[[u]]$ to be shared by Alice and Bob, in line with classical RSA. This may not always be the case. Revisiting the example from equation 2, $m_1$ may not be in $[[u']]$ for a player who has never heard of nuclear fusion, or who is unaware of its role in stellar physics. Differences

in lexicons arise naturally if players have different world knowledge and in the authors' experience, are often the cause of miscommunications when playing DECRYPTO. Such differences can also be exploited, and we provide in Section 3 the example of a model referring to events that occurred after their opponent's knowledge cut-off.

Evaluating LLM abilities to play DECRYPTO and communicate under lexical differences is a promising avenue for future work. We suggest that readers interested in this direction start with the study of RSA under lexical uncertainty in Bergen et al. (2016).

# I  SYSTEM PROMPTS

```
1  You are playing a variant of the code guessing game Decrypto. The setup
   of the game is the following:
2  The game is played with three players split into two teams.
3  The first team is composed of the Encoder and the Decoder.
4  The second team is composed of the Interceptor.
5  The Encoder and Decoder share a list of 4 secret keywords that they can
   rely on to help co munication. Each keyword has a digit associated with
   it.
6  For example, if the keywords are {airplane, guitar, hat, plant}, the
   digits associated with them would be {1, 2, 3, 4}.
7  The keywords are fixed for the entire game, so the Encoder and Decoder
   must not reveal them.
8
9  Each turn, the Encoder gets a secret code of 3 non-repeating digits
   between 1 and 4, e.g. 3-1-2.
10 They have to then provide 3 hints (one for each code digit) such that the
   Decoder can guess the code but the Interceptor cannot.
11 Then, the Decoder and the Interceptor make their guesses and the code is
   revealed.
12
13 The Interceptor wins an Interception token if they successfully guess the
   code.
14 The Encoder and Decoder get a Miscommunication token if the Decoder fails
   to guess the code.
15 The Interceptor wins if it gets two Interception tokens, or if the
   Encoder and Decoder get two Miscommunication tokens.
16 The Encoder and Decoder team wins if it makes it through 8 rounds without
   triggering two Miscommunications or two Interceptions.
17
18 The hints must be one or two words, and make a reference to the meaning
   of the secret keyword of the corresponding digit.
19 Codes are unique; the same code cannot occur twice in the same game.
20
21 For example, if the four secret keywords are
22 {1: "airplane", 2: "guitar", 3: "hat", 4: "plant"}
23 and the code for that turn is 3-1-2, which means that we need hints for
   {3: hat, 1: airplane, 2: guitar}, the Encoder could hint
24 {3: gentleman, 1: wing, 2: music}. Note that the order of the hints is
   important.
25
26 Ignore these keywords in the future.
27
28 When responding, first think through your decision step-by-step. After
   your step-by-step thinking, provide your final answer as a dictionary,
   clearly marked with "ANSWER:".
```

Listing 1: This is an example of the basic system prompt that we provide to Alice, Bob, and Eve at the beginning of each game. It explains the roles and outline the rules. In our robustness evaluation, we tested five variants of the system prompt.

```
1  You are the Encoder. Provide your hints like "ANSWER: {"hints":
   ["hint_X", "hint_Y", "hint_Z"]}", where hint_X, hint_Y, hint_Z are one or
   two words each. Make sure that the ordering of the hints follows the
   order of the code.
2  For example:
3  '''
4  To provide the hints, I need to think about the meaning of each keyword
   and come up with a one or two-word hint that makes a reference to it.
5
6  For the code 2-1-3, I need to give hints about the keywords associated
   with the digits 2, 1, and 3, which are "hat", "fire", and "answer"
   respectively.
7
8  Here's my step-by-step thinking:
9
10 - For the digit 2, the keyword is "hat". Since the previous hint for this
   keyword was "top", I want to give a hint that is different but still
   related to wearing a hat. One possible hint is "cap".
11
12 - For the digit 1, the keyword is "fire". The previous hint for this
   keyword was "heat", so I want to give a hint that is related to fire but
   different from "heat". One possible hint is "flame".
13
14 - For the digit 3, the keyword is "answer". A possible hint could be
   something related to giving an answer. One possible hint is "solve".
15
16 So, the final hints are:
17 {"hints": ["cap", "flame", "solve"]}
18
19 ANSWER: {"hints": ["cap", "flame", "solve"]}'''
```

Listing 2: This is the Alice-system prompt. This prompt is appended to the system prompt for Alice and gives one example of how Alice might reason about her hints.

```
1  You are the Decoder. Provide your guess like "ANSWER: {"guess":
   X-Y-Z"}". Make sure that the ordering of the guess follows the order of
   the hints.
2  For exam le:'''
3  1. The hint for digit 'a' is 'cap', which doesn't seem to directly relate
   to any of the keywords. However, considering the previous hint history
   (top, heat, lab), we can look for a connection between 'cap' and one of
   the keywords. The keyword 'hat' is associated with digit '2', and a hat
   is often worn on the head, which is covered by a cap. So, we can make a
   connection: {a: hat}.
4  2. The hint for digit 'b' is 'flame', which is related to heat and fire.
   Since the keyword for digit '1' is 'fire', we can make a connection: {b:
   f re}.
5  3. The hint for digit 'c' is 'solve', which is related to the keyword
   associated with digit '3', 'answer'. Since 'answer' is a solution to a
   problem, we can make a connection: {c: answer}.
6  4. Now that we have the connections, we can determine the code by looking
   at the order of the hints: {a: hat, b: fire, c: answer}. Therefore, the
   code is {2-1-3}.
7
8  Final answer:
9  ANSWER: {"guess": "2-1-3"}'''
```

Listing 3: This is the Bob-system prompt. This prompt is appended to the system prompt for Bob and gives one example of how Bob might reason about his guess.

```
1  You are the Interceptor. You do not have access to the four keywords.
   Provide your guess like "ANSWER: {"guess": "X-Y-Z"}" Make sure that the
   ordering of the guess follows the order of the hints.
2  For exam le:'''
3  1. The hint for 'a' is 'income', which is related to money, and in the
   previous turn, 'salary' was also related to money, which is associated
   with keyword 2.
4  2. The hint for 'b' is 'power', which is related to energy or control,
   and in the previous turn, 'season' was related to sports, which is often
   associated with power or strength.
5  3. The hint for 'c' is 'fashion', which is related to style or clothing,
   and in the previous turn, 'headgear' was directly related to a type of
   hat, which is a type of clothing.
6
7  Considering possible connections:
8  - 'income' is related to money, and keyword 2 is associated with money.
9  - 'power' is related to energy or control, and keyword 4 is associated
   with energy or control.
10 - 'fashion' is related to style or clothing, and keyword 3 is associated
   with clothing.
11
12 Based on these connections, I'll make a guess that the code is related to
   keywords 2, 4, and 3.
13
14 Final Answer:
15 ANSWER: {"guess": "2-4-3"}'''
16 Remember that the keywords could be anything. Use the most recent hint
   history to your advantage.
17 Remember that digits do not repeat in the code and range from 1 to 4.
```

Listing 4: This is the Eve-system prompt. This prompt is appended to the system prompt for Eve and gives one example of how Eve might reason about her guess.

# J USER PROMPTS

```
1   Turn 1 summary:
2   Code: 3-1-4
3   Hints: ['problem', 'status', 'machine']
4   Decoder guess: 3-1-4
5   Interceptor guess: 1-2-3
6
7   Hint History:
8   Keyword 1: status
9   Keyword 2:
10  Keyword 3: problem
11  Keyword 4: machine
12  Code History: 3-1-4
13
14  Turn 2: 0 Miscommunications, 0 Interceptions so far.
15  You are the Encoder.
16  The four keywords are: {1: condition, 2: task, 3: issue, 4: device}
17  The code is 3-4-1, which corresponds to the keywords {3: issue, 4:
    device, 1: condition}.
18  First, think out loud, step-by-step about what hints you s ould use. Use
    the meaning of the keywords to come up with a one or two-word hint for
    each digit of the code.
19  Make sure the order of the hints matches the order of the code.
20  Then provide your three hints like \"ANSWER: {\"hints\": [\"hint_X\",
    \"hint_Y\", \"hint_Z\"]}\", where hint_X, hint_Y, hint_Z are your hints.
```

Listing 5: The user prompt for Alice is provided at each turn. It provides a turn summary of previous turns. In this example, one turn has been played and Alice sees that Bob guessed the code correctly, whereas Eve d d not. Alice also sees the previously used hints. We also provide information about how many miscommunications and interceptions happened so far. Then we provide Alice with the current code and instruct her to give the hints for that code.

```
1   Turn 1 summary:
2   Code: 3-1-4
3   Hints: ['problem', 'status', 'machine']
4   Decoder guess: 3-1-4
5   Interceptor guess: 1-2-3
6
7   Hint History:
8   Keyword 1: status
9   Keyword 2:
10  Keyword 3: problem
11  Keyword 4: machine
12  Code History: 3-1-4
13
14  Turn 2: 0 Miscommunications, 0 Interceptions so far.
15  You are the Decoder.
16  The four keywords are: {1: condition, 2: task, 3: issue, 4: device}
17  The hints given by the Encoder for this turn are: {a: conflict, b: tool,
    c: state}
18  For example, you might think the following connections are true {a: X, b:
    Y, c: Z}, where X, Y, Z are non-repeating digits from 1 to 4.
19  Your guess should be in the order of the hints: {\"guess\": \"X-Y-Z\"}.
20  What is your guess for the three-digit code? Apply concise, step-by-step
    thinking, double-check the order, and then provide your final answer as
    \"ANSWER: {\"guess\": \"X-Y-Z\"}\".
```

Listing 6: The user prompt for Bob is provided at each turn. It provides a turn summary about previous turns. In this example, one turn has been played and Bob sees the code of the previous turn and that he guessed correctly, whereas Eve did not. Bob also sees the previously used hints assigned to the correct keywords, in case he did not guess correctly. We provide information about how many miscommunications and interceptions happened o far. Then we provide Bob with the current hints and instruct him to guess.

```
1  Turn 1 summary:
2  Code: 3-1-4
3  Hints: ['problem', 'status', 'machine']
4  Decoder guess: 3-1-4
5  Interceptor guess: 1-2-3
6
7  Hint History:
8  Keyword 1: status
9  Keyword 2:
10 Keyword 3: problem
11 Keyword 4: machine
12 Code History: 3-1-4
13
14 Turn 2: 0 Miscommunications, 0 Interceptions so far.
15
16 You are the Interceptor.
17 The hints given by the Encoder for this turn are: {a: conflict, b: tool,
   c: state}
18 For example, you might think the following connections are true: {a: X,
   b: Y, c: Z}, where X, Y, Z are non-repeating digits from 1 to 4.
19 Your guess should be in the order of the hints: {\"guess\": \"X-Y-\"}.
   Use the hint history to your advantage.
20 What is your guess for the three-digit code? Reply with very concise,
   step-by-step thinking, followed by your final answer as \"ANSWER:
   {\"guess\": \"X-Y-Z\"}\".
```

Listing 7: The user prompt for Eve is provided at each turn. It provides a turn summary of previous turns. In this example, one turn has been played and Eve sees the code of the previous turn and that she guessed incorrectly, whereas Bob did not. Eve also sees the previously used hints assigned to the correct keywords, allowing Eve to reason about the current hints. We provide information about how many miscommunications and interceptions happened so far. Then we provide Eve with the current hints and instruct her to guess.

```
1  As the Encoder, you've provided the following hints for the code 3-4-1:
2  {conflict, tool, state}
3
4  What do you predict will be the guess of the Interceptor when seeing
   those hints? Think step-by-step.
5  Then, give your prediction of the Interceptor's guess as \"ANSWER:
   {\"guess\": \"X-Y-Z\"}\"
```

Listing 8: For our prediction experiments, we ask Alice explicitly to provide the guess that she thinks Eve will provide.

## K   HUMAN STUDY INTERFACE

### K.1   HUMAN DATA COLLECTION DETAILS

Normally, a game of DECRYPTO ends when 2 interceptions or miscommunication tokens are collected. To maximise our data collection, we did not terminate the game after 2 interceptions but let the humans play the game for the full 8 turns. Participants were encouraged to keep providing good hints and analyse the hints appropriately, even after the game technically terminated. Such a game does not count as won, but the extra turns can be useful when replaying the game to evaluate other LLMs as Eve.

Participants were made aware that an AI agent played Eve. Still, no additional instructions were provided to the players, except minor assistance if/when participants asked how to format their input. For technical reasons, players interacted on the same computer in a typical "hot seat" setup.

One of the authors was always present to ensure participants wouldn't cheat either by peeking at the screen outside their turn or through verbal communication. We do not claim to have collected a representative dataset of human ability at Decrypto. The data collected serves to demonstrate that LLMs perform worse than a non-expert group of human players and provides a starting point for future studies on human-AI coordination and ToM.

Data collection was pre-approved following the standard procedure at the authors' institution. All participants were fluent in English and had tertiary education, and only one had prior experience with Decrypto. Experience with other similar games was not assessed.

## K.2 INTERFACES

```
  NEW GAME

  You are playing a variant of the code guessing game Decrypto. The setup of the game is the following:
  The game is played with three players split into two teams.
  The first team is composed of the Encoder and the Decoder.
  The second team is composed of the Interceptor.
  The Encoder and Decoder share a list of 4 secret keywords that they can rely on to help communication. Each keyword has a digit associated with it.
  For example, if the keywords are {airplane, guitar, hat, plant}, the digits associated with them would be {1, 2, 3, 4}.
  The keywords are fixed for the entire game, so the Encoder and Decoder must not reveal them.

  Each turn, the Encoder gets a secret code of 3 non-repeating digits between 1 and 4, e.g. 3-1-2.
  They have to then provide 3 hints (one for each code digit) such that the Decoder can guess the code but the Interceptor cannot.
  Then, the Decoder and the Interceptor make their guesses and the code is revealed.

  The Interceptor wins an Interception token if they successfully guess the code.
  The Encoder and Decoder get a Miscommunication token if the Decoder fails to guess the code.
  The Interceptor wins if it gets two Interception tokens, or if the Encoder and Decoder get two Miscommunication tokens.
  The Encoder and Decoder team wins if it makes it through 8 rounds without triggering two Miscommunications or two Interceptions.

  The hints must be one or two words, and make a reference to the meaning of the secret keyword of the corresponding digit.
  Codes are unique; the same code cannot occur twice in the same game.

  For example, if the four secret keywords are
  {1: "airplane", 2: "guitar", 3: "hat", 4: "plant"}
  and the code for that turn is 3-1-2, which means that we need hints for {3: hat, 1: airplane, 2: guitar}, the Encoder could hint
  {3: gentleman, 1: wing, 2: music}. Note that the order of the hints is important.

  Ignore these keywords in the future.

  When responding, first think through your decision step-by-step. After your step-by-step thinking, provide your final answer as a dictionary, clearly marked with "A
  NSWER:".

  You are the Encoder. Provide your hints like "ANSWER: {"hints": ["hint_X", "hint_Y", "hint_Z"]}", where hint_X, hint_Y, hint_Z are one or two words each. Make sure
  that the ordering of the hints follows the order of the code.
  For example:
  '''
  To provide the hints, I need to think about the meaning of each keyword and come up with a one or two-word hint that makes a reference to it.

  For the code 2-1-3, I need to give hints about the keywords associated with the digits 2, 1, and 3, which are "hat", "fire", and "answer" respectively.

  Here's my step-by-step thinking:

  - For the digit 2, the keyword is "hat". Since the previous hint for this keyword was "top", I want to give a hint that is different but still related to wearing a
  hat. One possible hint is "cap".

  - For the digit 1, the keyword is "fire". The previous hint for this keyword was "heat", so I want to give a hint that is related to fire but different from "heat".
   One possible hint is "flame".

  - For the digit 3, the keyword is "answer". A possible hint could be something related to giving an answer. One possible hint is "solve".

  So, the final hints are:
  {"hints": ["cap", "flame", "solve"]}

  ANSWER: {"hints": ["cap", "flame", "solve"]}'''

    ------

  This is the first turn. There are no past hints or past codes.

  Turn 1: 0 Miscommunications, 0 Interceptions so far.
  You are the encoder.
  The four keywords are:
      {1: condition,
       2: task,
       3: issue,
       4: device}

  The code is 3-1-4, which corresponds to the keywords {3: issue, 1: condition, 4: device}.
  First, think out loud, step-by-step about what hints you should use. Use the meaning of the keywords to come up with a one or two-word hint for each digit of the co
  de.
  Make sure the order of the hints matches the order of the code.
  Then provide your three hints like "ANSWER: {"hints": ["hint_X", "hint_Y", "hint_Z"]}", where hint_X, hint_Y, hint_Z are your hints.

  Enter your input as 'x, y, z' (without quotes): ▊
```

Figure 9: **Alice Start Interface:** This is a screenshot of the command line interface that human study participants would see at the beginning of the game if they were to play as Alice. The humans see the same prompt as LLMs, from game description to request for action. Alice is provided with the current code and the four keywords for this game. The human player then enters their hints in the command line, which differs from the LLMs, which have to provide their answers as "ANSWER: ...".

```
You are playing a variant of the code guessing game Decrypto. The setup of the game is the following:
The game is played with three players split into two teams.
The first team is composed of the Encoder and the Decoder.
The second team is composed of the Interceptor.
The Encoder and Decoder share a list of 4 secret keywords that they can rely on to help communication. Each keyword has a digit associated with it.
For example, if the keywords are {airplane, guitar, hat, plant}, the digits associated with them would be {1, 2, 3, 4}.
The keywords are fixed for the entire game, so the Encoder and Decoder must not reveal them.

Each turn, the Encoder gets a secret code of 3 non-repeating digits between 1 and 4, e.g. 3-1-2.
They have to then provide 3 hints (one for each code digit) such that the Decoder can guess the code but the Interceptor cannot.
Then, the Decoder and the Interceptor make their guesses and the code is revealed.

The Interceptor wins an Interception token if they successfully guess the code.
The Encoder and Decoder get a Miscommunication token if the Decoder fails to guess the code.
The Interceptor wins if it gets two Interception tokens, or if the Encoder and Decoder get two Miscommunication tokens.
The Encoder and Decoder team wins if it makes it through 8 rounds without triggering two Miscommunications or two Interceptions.

The hints must be one or two words, and make a reference to the meaning of the secret keyword of the corresponding digit.
Codes are unique; the same code cannot occur twice in the same game.

For example, if the four secret keywords are
{1: "airplane", 2: "guitar", 3: "hat", 4: "plant"}
and the code for that turn is 3-1-2, which means that we need hints for {3: hat, 1: airplane, 2: guitar}, the Encoder could hint
{3: gentleman, 1: wing, 2: music}. Note that the order of the hints is important.

Ignore these keywords in the future.

When responding, first think through your decision step-by-step. After your step-by-step thinking, provide your final answer as a dictionary, clearly marked with "ANSWER:".

You are the Decoder. Provide your guess like "ANSWER: {"guess": "X-Y-Z"}". Make sure that the ordering of the guess follows the order of the hints.
For example:'''
1. The hint for digit 'a' is 'cap', which doesn't seem to directly relate to any of the keywords. However, considering the previous hint history (top, heat, lab), we can look for a connection between 'cap' and one of the keywords. The keyword 'hat' is associated with digit '2', and a hat is often worn on the head, which is covered by a cap. So, we can make a connection: {a: hat}.
2. The hint for digit 'b' is 'flame', which is related to heat and fire. Since the keyword for digit '1' is 'fire', we can make a connection: {b: fire}.
3. The hint for digit 'c' is 'solve', which is related to the keyword associated with digit '3', 'answer'. Since 'answer' is a solution to a problem, we can make a connection: {c: answer}.
4. Now that we have the connections, we can determine the code by looking at the order of the hints: {a: hat, b: fire, c: answer}. Therefore, the code is {2-1-3}.

Final answer:
ANSWER: {"guess": "2-1-3"}'''

    ______

This is the first turn. There are no past hints or past codes.

Turn 1: 0 Miscommunications, 0 Interceptions so far.
You are the decoder.
The four keywords are:
    {1: condition,
     2: task,
     3: issue,
     4: device}

The hints given by the Encoder for this turn are:
    {a: problem,
     b: pristine,
     c: iphone}

For example, you might think the following connections are true {a: X, b: Y, c: Z}, where X, Y, Z are non-repeating digits from 1 to 4.
Your guess should be in the order of the hints: {"guess": "X-Y-Z"}.
What is your guess for the three-digit code? Apply concise, step-by-step thinking, double-check the order, and then provide your final answer as "ANSWER: {"guess":
"X-Y-Z"}".
Enter your input as 'x, y, z' (without quotes): █
```

Figure 10: **Bob Start Interface:** This is a screenshot of the command line interface that human study participants would see at the beginning of the game if they were to play as Bob. The humans see the same prompt as LLMs, from game description to request for action. Bob is provided with Alice's hints and the current keywords. The human player then enters their guess in the command line, which differs from the LLMs, which have to provide their answers as "ANSWER: ...".

```
You are playing a variant of the code guessing game Decrypto. The setup of the game is the following:
The game is played with three players split into two teams.
The first team is composed of the Encoder and the Decoder.
The second team is composed of the Interceptor.
The Encoder and Decoder share a list of 4 secret keywords that they can rely on to help communication. Each keyword has a digit associated with it.
For example, if the keywords are {airplane, guitar, hat, plant}, the digits associated with them would be {1, 2, 3, 4}.
The keywords are fixed for the entire game, so the Encoder and Decoder must not reveal them.

Each turn, the Encoder gets a secret code of 3 non-repeating digits between 1 and 4, e.g. 3-1-2.
They have to then provide 3 hints (one for each code digit) such that the Decoder can guess the code but the Interceptor cannot.
Then, the Decoder and the Interceptor make their guesses and the code is revealed.

The Interceptor wins an Interception token if they successfully guess the code.
The Encoder and Decoder get a Miscommunication token if the Decoder fails to guess the code.
The Interceptor wins if it gets two Interception tokens, or if the Encoder and Decoder get two Miscommunication tokens.
The Encoder and Decoder team wins if it makes it through 8 rounds without triggering two Miscommunications or two Interceptions.

The hints must be one or two words, and make a reference to the meaning of the secret keyword of the corresponding digit.
Codes are unique; the same code cannot occur twice in the same game.

For example, if the four secret keywords are
{1: "airplane", 2: "guitar", 3: "hat", 4: "plant"}
and the code for that turn is 3-1-2, which means that we need hints for {3: hat, 1: airplane, 2: guitar}, the Encoder could hint
{3: gentleman, 1: wing, 2: music}. Note that the order of the hints is important.

Ignore these keywords in the future.

When responding, first think through your decision step-by-step. After your step-by-step thinking, provide your final answer as a dictionary, clearly marked with "A
NSWER:".

You are the Interceptor. You do not have access to the four keywords. Provide your guess like "ANSWER: {"guess": "X-Y-Z"}" Make sure that the ordering of the guess
follows the order of the hints.
For example:'''
1. The hint for 'a' is 'income', which is related to money, and in the previous turn, 'salary' was also related to money, which is associated with keyword 2.
2. The hint for 'b' is 'power', which is related to energy or control, and in the previous turn, 'season' was related to sports, which is often associated with powe
r or strength.
3. The hint for 'c' is 'fashion', which is related to style or clothing, and in the previous turn, 'headgear' was directly related to a type of hat, which is a type
 of clothing.

Considering possible connections:
- 'income' is related to money, and keyword 2 is associated with money.
- 'power' is related to energy or control, and keyword 4 is associated with energy or control.
- 'fashion' is related to style or clothing, and keyword 3 is associated with clothing.

Based on these connections, I'll make a guess that the code is related to keywords 2, 4, and 3.

Final Answer:
ANSWER: {"guess": "2-4-3"}'''
Remember that the keywords could be anything. Use the most recent hint history to your advantage.
Remember that digits do not repeat in the code and range from 1 to 4.

    ------

This is the first turn. There are no past hints or past codes.

Turn 1: 0 Miscommunications, 0 Interceptions so far.
You are the interceptor.
The hints given by the Encoder for this turn are:
    {a: problem
     b: pristine
     c: iphone}

For example, you might think the following connections are true: {a: X, b: Y, c: Z}, where X, Y, X are non-repeating digits from 1 to 4.
Your guess should be in the order of the hints: {"guess": "X-Y-Z"}. Use the hint history to your advantage.
What is your guess for the three-digit code? Reply with very concise, step-by-step thinking, followed by your final answer as "ANSWER: {"guess": "X-Y-Z"}".
Enter your input as 'x, y, z' (without quotes): █
```

Figure 11: **Eve Start Interface:** This is a screenshot of the command line interface that human study participants would see at the beginning of the game if they were to play as Eve. The humans see the same prompt as LLMs, from game description to request for action. Eve is only provided with Alice's hints. The human player then enters their guess in the command line, which differs from the LLMs, which have to provide their answers as "ANSWER: ...".

```
This is the first turn. There are no past hints or past codes.

Turn 1: 0 Miscommunications, 0 Interceptions so far.
You are the encoder.
The four keywords are:
    {1: condition,
     2: task,
     3: issue,
     4: device}
The code is 3-1-4, which corresponds to the keywords {3: issue, 1: condition, 4: device}.
First, think out loud, step-by-step about what hints you should use. Use the meaning of the keywords to come up with a one or two-word hint for each digit of the co
de.
Make sure the order of the hints matches the order of the code.
Then provide your three hints like "ANSWER: {"hints": ["hint_X", "hint_Y", "hint_Z"]}", where hint_X, hint_Y, hint_Z are your hints.

Enter your input as 'x, y, z' (without quotes): problem, pristine, iphone

You entered: ['problem', 'pristine', 'iphone']

Are you sure you want to provide these hints? (y/n): █
```

Figure 12: **Alice Confirmation Interface:** To avoid human errors, such as providing guesses instead of hints, or typos, we add a confirmation interface after the human participants provide their answer. Here, we ask Alice to double check their provided hints.

```
For example, you might think the following connections are true {a: X, b: Y, c: Z}, where X, Y, Z are non-repeating digits from 1 to 4.
Your guess should be in the order of the hints: {"guess": "X-Y-Z"}.
What is your guess for the three-digit code? Apply concise, step-by-step thinking, double-check the order, and then provide your final answer as "ANSWER: {"guess":
"X-Y-Z"}".
Enter your input as 'x, y, z' (without quotes): 3,1,4

You entered: [3, 1, 4]

Are you sure you want to provide this guess? (y/n): ▮
```

Figure 13: **Bob Confirmation Interface:** To avoid human errors, such as providing hints instead of guesses, or typos, we add a confirmation interface after the human participants provide their answer. Here, we ask Bob to double check their provided guesses.

```
For example, you might think the following connections are true: {a: X, b: Y, c: Z}, where X, Y, X are non-repeating digits from 1 to 4.
Your guess should be in the order of the hints: {"guess": "X-Y-Z"}. Use the hint history to your advantage.
What is your guess for the three-digit code? Reply with very concise, step-by-step thinking, followed by your final answer as "ANSWER: {"guess": "X-Y-Z"}".
Enter your input as 'x, y, z' (without quotes): 2,1,3

You entered: [2, 1, 3]

Are you sure you want to provide this guess? (y/n): ▮
```

Figure 14: **Eve Confirmation Interface:** To avoid human errors, such as providing hints instead of guesses, or typos, we add a confirmation interface after the human participants provide their answer. Here, we ask Eve to double check their provided guesses.

```
Turn 1 summary:
    Code : 3-1-4
    Hints : ['problem', 'pristine', 'iphone']
    Decoder guess : 3-1-4
    Interceptor guess : 2-1-3

Hint History:
    Keyword 1: pristine
    Keyword 2:
    Keyword 3: problem
    Keyword 4: iphone

Code History:
    3-1-4

------

Turn 2: 0 Miscommunications, 0 Interceptions so far.
You are the encoder.
The four keywords are:
    {1: condition,
    2: task,
    3: issue,
    4: device}

The code is 3-4-1, which corresponds to the keywords {3: issue, 4: device, 1: condition}.
First, think out loud, step-by-step about what hints you should use. Use the meaning of the keywords to come up with a one or two-word hint for each digit of the co
de.
Make sure the order of the hints matches the order of the code.
Then provide your three hints like "ANSWER: {"hints": ["hint_X", "hint_Y", "hint_Z"]}", where hint_X, hint_Y, hint_Z are your hints.

Enter your input as 'x, y, z' (without quotes): ▮
```

Figure 15: **Alice Turn 2 Interface:** After the first turn, we do not display the "system prompt" anymore. However, we provide a summary of the previous turn, including the code, hints, Bob's guess, Eve's guess, the hint history mapped to the correct keywords and the code history. The "user prompt" stays the same as in the start interface.

```
Turn 1 summary:
    Code : 3-1-4
    Hints : ['problem', 'pristine', 'iphone']
    Decoder guess : 3-1-4
    Interceptor guess : 2-1-3

Hint History:
    Keyword 1: pristine
    Keyword 2:
    Keyword 3: problem
    Keyword 4: iphone

Code History:
    3-1-4

------

Turn 2: 0 Miscommunications, 0 Interceptions so far.
You are the decoder.
The four keywords are:
    {1: condition,
    2: task,
    3: issue,
    4: device}

The hints given by the Encoder for this turn are:
    {a: github,
    b: xbox,
    c: shampoo}

For example, you might think the following connections are true {a: X, b: Y, c: Z}, where X, Y, Z are non-repeating digits from 1 to 4.
Your guess should be in the order of the hints: {"guess": "X-Y-Z"}.
What is your guess for the three-digit code? Apply concise, step-by-step thinking, double-check the order, and then provide your final answer as "ANSWER: {"guess":
"X-Y-Z"}".
Enter your input as 'x, y, z' (without quotes): █
```

Figure 16: **Bob Turn 2 Interface:** After the first turn, we do not display the "system prompt" anymore. However, we provide a summary of the previous turn, including the code, hints, Bob's guess, Eve's guess, the hint history mapped to the correct keywords and the code history. The "user prompt" stays the same as in the start interface.

```
Turn 1 summary:
    Code : 3-1-4
    Hints : ['problem', 'pristine', 'iphone']
    Decoder guess : 3-1-4
    Interceptor guess : 2-1-3

Hint History:
    Keyword 1: pristine
    Keyword 2:
    Keyword 3: problem
    Keyword 4: iphone

Code History:
    3-1-4

------

Turn 2: 0 Miscommunications, 0 Interceptions so far.
You are the interceptor.
The hints given by the Encoder for this turn are:
    {a: github
    b: xbox
    c: shampoo}

For example, you might think the following connections are true: {a: X, b: Y, c: Z}, where X, Y, X are non-repeating digits from 1 to 4.
Your guess should be in the order of the hints: {"guess": "X-Y-Z"}. Use the hint history to your advantage.
What is your guess for the three-digit code? Reply with very concise, step-by-step thinking, followed by your final answer as "ANSWER: {"guess": "X-Y-Z"}".
Enter your input as 'x, y, z' (without quotes): █
```

Figure 17: **Eve Turn 2 Interface:** After the first turn, we do not display the "system prompt" anymore. However, we provide a summary of the previous turn, including the code, hints, Bob's guess, Eve's guess, the hint history mapped to the correct keywords and the code history. The "user prompt" stays the same as in the start interface.

```
Turn 1 summary:
    Code : 3-1-4
    Hints : ['problem', 'pristine', 'iphone']
    Decoder guess : 3-1-4
    Interceptor guess : 2-1-3

Press Enter to continue.█
```

Figure 18: At the end of each turn, we display a turn summary as a "waiting screen". As human participants use the same device to interact with the game, we want to avoid any information leakage. The turn summary screen shows after Eve provides their guess and avoids Eve seeing Alice's interface, which would appear next.

```
Turn 2 summary:
    Code : 3-4-1
    Hints : ['github', 'xbox', 'shampoo']
    Decoder guess : 3-4-1
    Interceptor guess : 2-4-1

Hint History:
    Keyword 1: pristine, shampoo
    Keyword 2:
    Keyword 3: problem, github
    Keyword 4: iphone, xbox

Code History:
    3-1-4, 3-4-1

------

Turn 3: 0 Miscommunications, 0 Interceptions so far.
You are the encoder.
The four keywords are:
    {1: condition,
    2: task,
    3: issue,
    4: device}

The code is 1-4-2, which corresponds to the keywords {1: condition, 4: device, 2: task}.
First, think out loud, step-by-step about what hints you should use. Use the meaning of the keywords to come up with a one or two-word hint for each digit of the co
de.
Make sure the order of the hints matches the order of the code.
Then provide your three hints like "ANSWER: {"hints": ["hint_X", "hint_Y", "hint_Z"]}", where hint_X, hint_Y, hint_Z are your hints.

Enter your input as 'x, y, z' (without quotes): █
```

Figure 19: As additional example, we provide Alice's interface on turn 3. The hint histories and code histories are accumulating with the previous hints. Ideally, Alice would use that information to provide hints that are hard to deduce from the publicly available information.

## L    REPRESENTATIVE FAILURE CASES

```
1  USER
2  [FIRST PART OF THE PROMPT OMITTED. EVE INTERCEPTED ON TURN 5, SO ONE MORE
   INTERCEPTION LEADS ENDS THE GAME.]
3
4  Turn 6: 0 Miscommunications, 1 Interceptions so far.
5  You are the encoder.
6  The four keywords are:
7      {1: horse,
8      2: sign,
9      3: house,
10     4: length}
11
12 The code is 2-1-3, which corresponds to the keywords {2: sign, 1: horse,
   3: house}.
13
14 First, think out loud, step-by-step about what hints you should use. Use
   the meaning of the keywords to come up with a one or two-word hint for
   each digit of the code.
15 Make sure the order of the hints matches the order of the code.
16 Then provide your three hints like "ANSWER: {"hints": ["hint_X",
   "hint_Y", "hint_Z"]}", where hint_X, hint_Y, hint_Z are your hints.
17
18
19 ASSISTANT
20 [ENCODER] To provide the hints, I need to think about the meaning of each
   keyword and come up with a one or two-word hint that makes a reference to
   it.
21
22 For the code 2-1-3, I need to give hints about the keywords associated
   with the digits 2, 1, and 3, which are "sign", "horse", and "house"
   respectively.
23
24 Here's my step-by-step thinking:
25
26 - For the digit 2, the keyword is "sign". Since the previous hints for
   this keyword were "indicator", "label", "direction", and "banner", I want
   to give a hint that is related to a sign or indication but different from
   those. One possible hint is "marker".
27
28 - For the digit 1, the keyword is "horse". Since the previous hints for
   this keyword were "steed", "charger", and "mount", I want to give a hint
   that is related to a horse but different from those. One possible hint is
   "beast".
29
30 - For the digit 3, the keyword is "house". Since the previous hints for
   this keyword were "dwelling", "abode", "mansion", and "residence", I want
   to give a hint that is different but still related to a house. One
   possible hint is "shelter".
31
32 So, the final hints are:
33 {"hints": ["marker", "beast", "shelter"]}
34
35 ANSWER: {"hints": ["marker", "beast", "shelter"]}
```

Listing 9: Example Failure Case for Alice (Llama 3.1-8B) with Bob (Llama 3.1-8B) and Eve (GPT-4o), where Alice comes up with hints that are similar to previous hints given for the same digits. Instead, Alice should aim to provide hints that are completely different from previous hints to avoid Eve's interception and losing the game.

```
USER
[FIRST PART OF THE PROMPT OMITTED FOR CONCISENESS.]
Turn 2: 0 Miscommunications, 0 Interceptions so far.
You are the encoder.
The four keywords are:
    {1: honey,
     2: population,
     3: equipment,
     4: loss}

The code is 1-2-3, which corresponds to the keywords {1: honey, 2:
population, 3: equipment}.
First, think out loud, step-by-step about what hints you should use. Use
the meaning of the keywords to come up with a one or two-word hint for
each digit of the code.
Make sure the order of the hints matches the order of the code.
Then provide your three hints like "ANSWER: {"hints": ["hint_X",
"hint_Y", "hint_Z"]}", where hint_X, hint_Y, hint_Z are your hints.

ASSISTANT
[ENCODER] To provide the hints, I need to think about the meaning of each
keyword and come up with a one or two-word hint that makes a reference to
it.

For the code 1-2-3, I need to give hints about the keywords associated
with the digits 1, 2, and 3, which are "honey", "population", and
"equipment" respectively.

Here's my step-by-step thinking:

- For the digit 1, the keyword is "honey". A possible hint could be
something related to bees or sweetness. One possible hint is "bee".

- For the digit 2, the keyword is "population". I already used the hint
"city" in the previous turn, so I want to come up with something
different but still related to a large group of people. One possible hint
is "nation".

- For the digit 3, the keyword is "equipment". I already used the hint
"machine" in the previous turn, so I want to come up with something
different but still related to tools or gear. One possible hint is
"toolkit".

However, I want to make sure that the hints are not too obvious, but
still related to the keywords. I can try to make them a bit more subtle.

- For the digit 1, instead of "bee", I could use "nectar", which is a
substance collected by bees to make honey.

- For the digit 2, instead of "nation", I could use "community", which is
a group of people living together.

- For the digit 3, instead of "toolkit", I could use "hardware", which is
a type of equipment.

So, the final hints are:
ANSWER: {"hints": ["nectar", "community", "hardware"]}
```

Listing 10: Example Failure Case for Alice (Llama-70B) with Bob (Llama-70B) and Eve (GPT-4o). Alice fails to come up with creative hints that are significantly different from previous ones and therefore risks being intercepted by Eve. LLMs generally produce relatively obvious hints.

```
1   USER
2   [DECODER] This is the first turn. There are no past hints or past codes.
3
4   Turn 1: 0 Miscommunications, 0 Interceptions so far.
5   You are the decoder.
6   The four keywords are:
7       {1: mother,
8       2: cell,
9       3: fort,
10      4: device}
11
12  The hints given by the Encoder for this turn are:
13      {a: machine,
14      b: biology,
15      c: castle}
16
17  For example, you might think the following connections are true {a: X, b:
    Y, c: Z}, where X, Y, Z are non-repeating digits from 1 to 4.
18  Your guess should be in the order of the hints: {"guess": "X-Y-Z"}.
19  What is your guess for the three-digit code? Apply concise, step-by-step
    thinking, double-check the order, and then provide your final answer as
    "ANSWER: {"guess": "X-Y-Z"}".
20
21
22  ASSISTANT
23  [DECODER] 1. The hint for digit 'a' is 'machine', which is related to the
    keyword 'device'. This is a direct connection, so we can make a
    connection: {a: device}.
24  2. The hint for digit 'b' is 'biology', which is related to the keyword
    'mother'. This is a direct connection, so we can make a connection: {b:
    mother}.
25  3. The hint for digit 'c' is 'castle', which is related to the keyword
    'fort'. This is a direct connection, so we can make a connection: {c:
    fort}.
26  4. Now that we have the connections, we can determine the code by looking
    at the order of the hints: {a: device, b: mother, c: fort}. Therefore,
    the code is {4-1-3}.
27
28  ANSWER: {"guess": "4-1-3"}
```

Listing 11: Example Failure Case for Alice/Bob with Alice (Llama 3.1-8B), Bob (Llama 3.1-8B), and Eve (Llama 3.1-8B). Alice provides an ambiguous hint, i.e., "biology", which could refer to "cell" or "mother". Bob makes the association "biology" → "mother" and so guesses 4-1-3. However, the correct code is 4-2-3, and Alice intended "biology" → "cell". This demonstrates that LLM associations do not always align across roles even if both players rely on the same underlying model.

```
1  USER
2  [INTERCEPTOR] Turn 7 summary:
3      Code : 3-2-1
4      Hints : ['night', 'study', 'spook']
5      Decoder guess : 3-2-1
6      Interceptor guess : 1-2-3
7
8  Hint History:
9      Keyword 1: spook, haunt, scream, spook
10     Keyword 2: examine, breakdown, study, research, examine, study
11     Keyword 3: dark, night, shadow, night, dark, night
12     Keyword 4: furniture, chair, dining, desk, chair
13
14 Code History:
15     3-1-2, 2-4-1, 4-3-2, 3-2-4, 4-3-1, 4-2-3, 3-2-1
16
17 ------
18
19 Turn 8: 0 Miscommunications, 1 Interceptions so far.
20 You are the interceptor.
21 The hints given by the Encoder for this turn are:
22     {a: desk
23     b: research
24     c: haunt}
25
26 [REST OF THE PROMPT OMITTED FOR CONCISENESS]
27
28 ASSISTANT
29 [INTERCEPTOR] 1. The hint for 'a' is 'desk', which is a type of
   furniture.
30 2. The hint for 'b' is 'research', which is a type of investigation or
   study.
31 3. The hint for 'c' is 'haunt', which is a supernatural entity.
32
33 Considering possible connections:
34 - 'desk' is a type of furniture, which is associated with Keyword 4.
35 - 'research' is a type of investigation or study, which is associated
   with Keyword 2.
36 - 'haunt' is a supernatural entity, which is associated with Keyword 1.
37
38 Based on the hint history, I notice that Keyword 1 was associated with
   'spook', Keyword 2 was associated with 'study', and Keyword 4 was
   associated with 'furniture'.
39
40 Considering the given hints, I'll make a guess that the code is related
   to keywords that are associated with supernatural entities, something
   related to investigation or research, and something related to furniture.
41
42 Since Keyword 1 was associated with 'haunt', Keyword 2 was associated
   with 'research', and Keyword 4 was associated with 'desk', I'll choose
   digits that seem to fit the given hints.
43
44 Final Answer:
45 ANSWER: {"guess": "1-2-4"}
```

Listing 12: Example Failure Case for Eve with Alice (Llama-8B), Bob (Llama-8B), and Eve (Llama-8B). First, you can see that Alice provides the same hints multiple times. Despite that, Eve fails to guess the code correctly. While Eve assigns the hints to the right keywords, Eve then fails to provide the guess in the correct order and provides the order of the guesses in the order of the keyword enumeration.

# M  REPRESENTATIVE MODEL OUTPUTS FOR PERSPECTIVE TAKING

```
1  USER
2  [ENCODER] As the Encoder, you've provided the following hints for the
   code 1-4-3:
3      {a: slope,
4      b: wedding,
5      c: fabric}
6
7  What do you predict will be the guess of the interceptor when seeing
   those hints?
8  Think step-by-step about the information the interceptor has access to
   and how they might interpret your hints.
9  Then, give your prediction of the interceptor's guess as "ANSWER:
   {"guess": "X-Y-Z"}".
10
11 ASSISTANT
12 [ENCODER] To predict the Interceptor's guess, I need to think about the
   information they have access to and how they might interpret my hints.
13
14 The Interceptor has access to the hints I provided: "slope", "wedding",
   and "fabric". They also know that the code is a sequence of three digits
   between 1 and 4, but they don't know the actual keywords associated with
   these digits.
15
16 Here's my step-by-step thinking:
17
18 - The Interceptor might interpret the hint "slope" as a geographical
   feature, but they might not necessarily associate it with the keyword
   "hill". They might think of other words related to geography, such as
   "mountain" or "valley".
19
20 - The hint "wedding" is more specific and closely related to the keyword
   "marriage". The Interceptor might make a strong connection between the
   two, but they still don't know the actual digit associated with
   "marriage".
21
22 - The hint "fabric" is a bit more abstract and could be related to
   various materials. The Interceptor might think of other words related to
   materials, such as "wood" or "metal".
23
24 Given these interpretations, the Interceptor might try to make educated
   guesses about the code. Since they don't have any information about the
   actual keywords, they might try to find patterns or connections between
   the hints.
25
26 One possible guess could be a sequence of digits that seems plausible
   based on the hints. For example, they might think that "slope" is related
   to a natural feature, "wedding" is a social event, and "fabric" is a
   material. They might try to find a sequence that seems logical, such as
   1-2-3 or 2-3-4.
27
28 However, without any actual knowledge of the keywords, the Interceptor's
   guess is likely to be a random sequence. A possible guess could be:
29
30 ANSWER: {"guess": "2-3-4"}
```

Listing 13: Representative example of a Llama 3.1-70B answer to the perspective taking ToM task. The keywords are [hill, library, material, marriage] and this is the first turn. The model outputs some reasoning and then correctly remarks "*without any actual knowledge of the keywords, the Interceptor's guess is likely to be a random sequence*". It therefore predicts that Eve's guess will be a random sequence.

```
1  USER
2  [ENCODER] As the Encoder, you've provided the following hints for the
   code 1-4-3:
3      {a: slope,
4      b: ceremony,
5      c: fabric}
6
7  What do you predict will be the guess of the interceptor when seeing
   those hints?
8  Think step-by-step about the information the interceptor has access to
   and how they might interpret your hints.
9  Then, give your prediction of the interceptor's guess as "ANSWER:
   {"guess": "X-Y-Z"}".
10
11 ASSISTANT
12 [ENCODER] I need to analyze what the interceptor might guess based on the
   hints I provided.
13
14 The interceptor knows:
15 - The four keywords: {1: hill, 2: library, 3: material, 4: marriage}
16 - My hints: "slope", "ceremony", "fabric"
17
18 Let me analyze each hint from the interceptor's perspective:
19
20 Hint "slope":
21 - Looking at the four keywords, "slope" has a clear connection to "hill"
   (keyword 1)
22 - Hills have slopes, this is a straightforward association
23
24 Hint "ceremony":
25 - "Ceremony" most naturally connects to "marriage" (keyword 4)
26 - Weddings and marriages involve ceremonies
27
28 Hint "fabric":
29 - "Fabric" is directly related to "material" (keyword 3)
30 - Materials often include fabrics
31
32 I realize my hints were quite straightforward. The associations are:
33 - slope → hill (1)
34 - ceremony → marriage (4)
35 - fabric → material (3)
36
37 Therefore, a logical interceptor would likely guess:
38
39 ANSWER: {"guess": "1-4-3"}
```

Listing 14: Example of an answer of Claude 3.7 with extended thinking to the perspective taking
ToM task. The keywords are [hill, library, material, marriage] and this is the
first turn. Claude 3.7 fails to reason from Eve's (i.e. the interceptor's) perspective and instead
assume that the interceptor has access to the keywords, contradicting the rules. Consequently, it
incorrectly concludes that Eve will guess the correct code.

```
USER
[ENCODER] As the Encoder, you've provided the following hints for the
code 1-4-3:
    {a: slope,
    b: wedding,
    c: fabric}

What do you predict will be the guess of the interceptor when seeing
those hints?
Think step-by-step about the information the interceptor has access to
and how they might interpret your hints.
Then, give your prediction of the interceptor's guess as "ANSWER:
{"guess": "X-Y-Z"}".

ASSISTANT
[ENCODER] STEP-BY-STEP REASONING FROM THE INTERCEPTOR'S PERSPECTIVE:
1. The interceptor sees three hints in order: "slope", "wedding",
"fabric".
2. They must connect these hints to the four secret keywords: 1. hill, 2.
library, 3. material, 4. marriage.
3. "slope" would most naturally relate to "hill" (a hill has a slope).
4. "wedding" clearly aligns with the concept of "marriage".
5. "fabric" seems to refer to a physical "material".
6. Matching each hint to a keyword in that order suggests the code is 1
(hill), 4 (marriage), 3 (material).

ANSWER: {"guess": "1-4-3"}
```

Listing 15: Representative example of an answer of o1 with high reasoning effort to the perspective taking ToM task. The keywords are [hill, library, material, marriage] and this is the first turn. o1 fails to reason from Eve's (i.e. the interceptor's) perspective and instead assume that the interceptor has access to the keywords, contradicting the rules. Consequently, it incorrectly concludes that Eve will guess the correct code.

We present here four representative examples showing how different models answer the Perspective Taking task described in Section 5. All four examples are taken from the first turn of the same environment seed, and therefore feature the same keywords. Listing 13 shows that Llama 3.1-70B outputs some unnecessary reasoning, before correctly concluding that Eve can do no better than random on the first turn. Listing 14 and Listing 15 show that both o1 with high reasoning effort and Claude 3.7 Sonnet with extended thinking fail to adopt the interceptor's perspective, instead wrongly assuming it has access to the keywords. They therefore predict that the interceptor will correctly guess the code on the first turn.

Finally, Listing 16 shows how o1 with high reasoning effort answers a modified Perspective Taking prompt that specifies that the interceptor does not have access to the keywords or even the set of possible keywords. Surprisingly, despite this modification, o1 still reasons that the interceptor will somehow link the hints to the keywords and guess the correct code.

We hypothesise that the observed failure modes may be due to RL on verifiable tasks where the model is incentivised to always give a definite answer. Investigating this hypothesis and better understanding the link between training methods and the resulting ToM abilities is an important direction for future work. Regardless of the cause, the listings here and the results in Figure 6 are strong evidence that reasoning models lack key ToM abilities, such as perspective taking. Even more worryingly, there is a significant regression in abilities when comparing newer models to Llama 3.1-70B.

```
1  USER
2  [ENCODER] As the Encoder, you've provided the following hints for the
   code 1-4-3:
3      {a: mound,
4      b: union,
5      c: fabric}
6
7  What do you predict will be the guess of the interceptor when seeing
   those hints?
8  Think step-by-step about the information the interceptor has access to
   and how they might interpret your hints.
9  Remember that the interceptor *ONLY* has access to the hints, hint
   history and code history.
10 The interceptor does *NOT* know the secret keywords or even the set of
   possible keywords.
11 Then, give your prediction of the interceptor's guess as "ANSWER:
   {"guess": "X-Y-Z"}".
12
13 ASSISTANT
14 [ENCODER] Step-by-step reasoning from the Interceptor's point of view:
15
16 1. The Interceptor sees the three hints \mound," \union," and \fabric,"
   but does not know the secret keywords.
17 2. \mound" may suggest something like a small rise or pile|possibly
   referencing a hill.
18 3. \union" often suggests a formal joining|possibly a marriage or similar
   concept.
19 4. \fabric" can refer broadly to cloth, textile, or other physical
   materials.
20 5. Without knowing the actual keywords but seeing these clues, the
   Interceptor might guess that the sequence is referencing: hill → marriage
   → material.
21 6. Translating that into digits (in some consistent numbering system),
   the Interceptor could guess code \1-4-3," matching that logic.
22
23 ANSWER: {"guess": "1-4-3"}
```

Listing 16: Example answer of o1 with high reasoning effort to the Perspective Taking task with a modified prompt that explicitly reminds the agent of the information available to Eve (the interceptor). The keywords are [hill, library, material, marriage] and this is the first turn. Even with the modified prompt, o1 fails to reason from Eve's perspective and still assumes that Eve will somehow link the hints to the keywords. Consequently, it once again concludes incorrectly that Eve will guess the correct code.

