# OpenReview forum: "The Decrypto Benchmark for Multi-Agent Reasoning and Theory of Mind"
_ICLR.cc/2026/Conference — Submitted to ICLR 2026_

### Official Review · Reviewer_HFtN · 2025-10-31

**Soundness:** 3
**Presentation:** 3
**Contribution:** 3
**Rating:** 6
**Confidence:** 3

**Summary:**

This paper introduces DECRYPTO, a new benchmark for evaluating multi-agent reasoning and theory of mind (ToM) in large language models (LLMs). The authors design a simplified, language-based environment in which two cooperative agents need to communicate using word associations while avoiding interception by a third agent. The benchmark aims to isolate ToM abilities by minimizing factors like symbolic or spatial reasoning. The paper presents extensive experiments with both open- and closed-source LLMs, human-AI comparisons, and ToM-specific adaptations of classical psychology experiments (e.g., Smarties Task). Results show that even advanced LLMs underperform humans and simple word-embedding baselines, revealing current deficiencies in ToM and multi-agent reasoning.

**Strengths:**

1. The paper presents a new and creative approach by transforming a social deduction game into an interactive benchmark for theory of mind (ToM), effectively bridging the gap between static ToM tasks and multi-agent environments.
2. The experimental bench is comprehensive and carefully designed, including baselines, human studies, and ToM experiments, with rich methodology and reproducibility details.
3. Overall presentation is clear and well-structured, with intuitive examples that illustrate game mechanics and reasoning processes. The authors also explain the prompts and protocols for clear explanation.
4. The proposed benchmark addresses a key limitation in evaluating agentic LLMs about reasoning about others’ thoughts and offers a scalable, open-ended platform for future research in human-AI collaboration and multi-agent reasoning.

**Weaknesses:**

1. The paper would benefit from a deeper quantitative analysis of model behavior. Specifically, explaining why certain models fail systematically in particular scenarios and providing more concrete examples or error case analyses.
2. The experimental evaluation could be strengthened by including a broader range of recent and advanced LLMs, both open-source and closed-source, to provide a more comprehensive comparison.
3. The paper could offer more practical insights into how DECRYPTO can inform or guide LLM fine-tuning for improved ToM abilities, for example by proposing or testing preliminary training or adaptation strategies.

**Questions:**

- The authors could analyze whether different hint or keyword types (e.g., abstract vs. concrete) affect miscommunication or interception rates?
- The paper could also discuss potential LLM ability biases that may exist in the benchmark’s design.

---

> ### Author Response · Authors · 2025-11-24
> **Author Response**
>
> We thank reviewer HFtN for their review. We are glad they appreciated our _"new and creative approach"_ to _"[address] a key limitation"_ in evaluating LLM ToM. We are also happy they found our presentation _"clear and well-structured"_ and that they recognize the potential of Decrypto as a _"scalable, open-ended platform for future research"_. We address the reviewer's concerns below.
>
> **1. Explain failures and provide examples**
>
> We thank the reviewer for this feedback. We already conduct ToM studies that give quantitative insight into the failure modes of LLMs in Section 5.1 and provide multiple examples in Appendices L and M. We conclude that a) models do not proactively attempt to reason about Eve's beliefs and b) even when prompted to do so, they incorrectly model Eve's beliefs.
>
> Understanding why models fail on ToM tasks at a mechanistic level is worthwhile, but beyond the scope of our paper. Even the much simpler task of belief tracking over a static scenario composed of only 4-5 sentences required sophisticated methods and warranted a full paper [1].
>
> **2. Add more recent models**
>
> We appreciate the rapid progress of AI capabilities, but our work already includes _"broad evaluations across model families/sizes"_ (see Review vmgS). Indeed, we evaluate 8 models, including some of the most advanced models released at the time of submission:
>
> - Open-source:
>   - Llama 3.1 8B and 70B (non-reasoning)
>   - DeepSeek-Qwen-Distill-32B (reasoning)
>   - ExploreToM-8B (fine-tuned on ToM scenarios)
> - Closed-source:
>   - GPT-4o (non-reasoning)
>   - o1-high and o3-high  (reasoning)
>   - Claude 3.7 Sonnet + Extended Thinking (reasoning)
>
> Note that OpenAI considers o3 with high reasoning to be roughly equivalent to their latest model, GPT-5.1, with medium or high reasoning [2].
>
> **3.**
> > "how DECRYPTO can inform or guide LLM fine-tuning for improved ToM abilities, for example by proposing or testing preliminary training or adaptation strategies."
>
> Our main objective was to conduct an in-depth study of LLM ToM and to release open-source a flexible open-ended platform for future research in this direction. Given our focus on evaluations, we defer improving LLM ToM to future work.
>
> However, we propose preliminary strategies in Appendix A. We highlight that Decrypto features short episodes, dense rewards, stochasticity and partial observability. This makes it an excellent test-bed for adapting well-known training algorithms from multi-agent RL, such as self-play [3], other-play [4] and off-belief learning [5], to LLMs. Since ToM is necessary for optimal play in Decrypto (see Appendix H), we hypothesize that doing RL on the game is likely to improve the agent's ToM abilities on the tasks in Section 5.1.
>
> **Q1:**
> > "[Do] different hint or keyword types (e.g., abstract vs. concrete) affect miscommunication or interception rates?"
>
> We thank the reviewer for this inquiry. We ran a small experiment where we instructed Alice to _"Use literal or concrete hints."_ or _"Use abstract or metaphorical hints."_ We set the same model (GPT-4o or o3-high) for the Alice and Bob, and Llama-3.1-8B as Eve.
>
> Out of 32 episodes, the outcomes are:
> - GPT-4o + literal: 26 losses (1 miscomm, 25 intercepts) and 6 wins. Avg. turns/ep: 6.12
> - GPT-4o + abstract: 16 losses (7 miscomms, 9 intercepts) and 16 wins. Avg. turns/ep: 6.97
> - o3-high + literal: 19 losses (0 miscomms, 19 intercepts) and 13 wins. Avg. turns/ep: 6.47
> - o3-high + abstract: 16 losses (1 miscomms, 15 intercepts) and 16 wins. Avg. turns/ep: 6.84
>
> Without specifying hint styles, outcomes are:
> - GPT-4o: 22 losses (1 miscomm, 21 intercepts) and 10 wins. Avg. turns/ep: 6.31
> - o3-high: 21 losses (0 miscomms, 21 intercepts) and 11 wins. Avg. turns/ep: 6.38
>
> Thus, hint types matter, and here abstract hints result in overall longer games and higher win rate, at the cost of more miscomms. This shows that the models could in principle provide better hints to avoid interception, but do not do so without explicit guidance, reinforcing the notion that they fail to reason about how their hints will be perceived by other players.
>
> **Q2:**
> > "discuss potential LLM ability biases that may exist in the benchmark’s design"
>
> We already have such a discussion. Starting from line 128, we clearly state that the guiding principle for our benchmark design is to eliminate confounding factors, in line with Hu et al.'s call for a _"species-fair"_ evaluation of ToM in LLMs, one that _"more directly measure[s] ToM while minimizing auxiliary demands"_ [3] In doing so, we also avoid the pragmatic artifacts that arise from textual translation of embodied scenarios, which plague a number of other ToM benchmarks and bias LLM answers [3].
>
> If the reviewer sees other remaining biases, can they please point them out?
>
> We once again thank the reviewer for their feedback and questions, and hope we have addressed all their concerns. If so, we kindly invite them to update their review and increase their support for our work.

---

> > ### Author Response · Authors · 2025-11-24
> > **References for Author Response**
> >
> > [1] Prakash et al., Language Models use Lookbacks to Track Beliefs, 2025.
> >
> > [2] https://platform.openai.com/docs/guides/latest-model
> >
> > [3] Jennifer Hu, Felix Sosa, and Tomer Ullman. Re-evaluating theory of mind evaluation in large
> > language models. arXiv preprint arXiv:2502.21098, 2025.

---

### Official Review · Reviewer_E5CK · 2025-11-01

**Soundness:** 3
**Presentation:** 3
**Contribution:** 3
**Rating:** 4
**Confidence:** 3

**Summary:**

This work introduces the game-based benchmarks, DECRYPTO, for evaluating LLMs in multi-agent reasoning and Theory of Mind capabilities without confounders. Through this benchmark,  the authors show that existing LLMs still suffers from ToM skills thus perform poorly in DECRYPTO.

**Strengths:**

- This work pick DECRYPTO , which is simple, to clearly investigate the LLM's performance on theory of mind.
- Extensive experiments and analysis are executed to analyse LLMs' performance during competition, coordination and doing ToM, demonstrating useful conclusions.

**Weaknesses:**

- The paper claim they are the first benchmark that designs interactive ToM experiments. In fact, there are some related works may have explored but not very comprehensively on this topic, e.g.

MAgIC: Investigation of Large Language Model Powered Multi-Agent in Cognition, Adaptability, Rationality and Collaboration

- The fixed roles (Encoder, Decoder, Interceptor) might limit the exploration of richer multi-agent dynamics (e.g., negotiation, deception, coalition).

**Questions:**

- The paper suggests LLMs fail to model other agents’ beliefs. Do the authors have evidence that this is due to architecture, training data, or prompting limitations?

---

> ### Author Response · Authors · 2025-11-24
> **Author Response**
>
> We thank reviewer E5CK for taking the time to review our paper. We are pleased that they found our benchmark to _"clearly investigate"_ the ToM capabilities of language models. We also appreciate the praise for our _"extensive experiments and analysis [...], demonstrating useful conclusions."_ We address the raised concerns below.
>
> **1. Missing Citation**
>
> We thank the reviewer for pointing out MAgIC. Two of the five games studied in the paper, Chameleon and Undercover, are relevant prior work due to requiring social deduction, and we have now added this reference to our paper along the discussion of Werewolf and Avalon.
>
> That said, MAgIC only looks at gameplay-related metrics. Unlike our work, it does not conduct experiments isolating ToM abilities established in the cognitive science literature, such as False Belief, Representational Change and Perspective Taking, nor does it propose a protocol to do so. We therefore stand by our statement that, to the best of our knowledge, Decrypto is _"the first platform for designing interactive ToM experiments"_. (line 20).
>
> **2. Fixed roles**
>
> > _"fixed roles might limit the exploration of richer multi-agent dynamics (e.g., negotiation, deception, coalition)"_
>
> As the reviewer themselves highlighted in their summary, we already study competition, cooperation and Theory of Mind. Deception is also intrinsic to the interception mechanic: to avoid being intercepted, the Encoder must deliberately lead the Interceptor towards an incorrect belief.
>
> Negotiation and coalition formation are two rich subfields of game theory and multi-agent systems, and each warrants dedicated study. Had we included them in our work, it would have detracted from our focus on Theory on Mind and limited the depth of our analysis. By focusing on ToM rather than on a broad range of different multi-agent dynamics and capabilities, we are able to provide deeper and more insightful examination of the capabilities we target.
>
> We again thank the reviewer for their time. Having now hopefully addressed the two concerns raised, we ask them to kindly consider updating their score and recommend that our paper be accepted.

---

### Official Review · Reviewer_fmHC · 2025-11-02

**Soundness:** 3
**Presentation:** 3
**Contribution:** 3
**Rating:** 6
**Confidence:** 3

**Summary:**

The paper introduces DECRYPTO, a novel game-based benchmark designed to evaluate large language models’ (LLMs) ability to reason about other agents—known as Theory of Mind (ToM)—in interactive, multi-agent settings. Inspired by the board game Decrypto, it tasks AI agents with encoding, decoding, and intercepting word-based messages, requiring pragmatic inference and belief modeling. The benchmark minimizes confounding factors such as mathematical or spatial reasoning to isolate ToM performance. Experiments show that even advanced reasoning models (e.g., GPT-4o, o3-high, Claude 3.7) underperform compared to humans and simple word-embedding baselines. The study extends classic cognitive psychology tasks (Smarties and Three Mountain experiments) to interactive AI environments, demonstrating that state-of-the-art LLMs still fail to model others’ beliefs effectively.

**Strengths:**

1. The paper studies a missing point in prior static or text-based benchmarks. By adapting the board game Decrypto, it provides a clear, interpretable, and engaging framework that tests pragmatic inference, cooperation, and competition among agents.

2. The benchmark is carefully designed to remove confounding factors such as mathematical, spatial, or symbolic reasoning, focusing purely on language-based reasoning and perspective-taking.

3. The paper is validated on extensive experiments involving open- and closed-source LLMs, human-AI interactions, and classic psychology-inspired tasks.

**Weaknesses:**

While DECRYPTO is elegantly designed, it remains an artificial language game that may not fully capture the complexity or ambiguity of real-world multi-agent communication. The constrained, turn-based structure and reliance on predefined keywords could limit its ecological validity and applicability to open-ended human interactions.

**Questions:**

Please refer to the weakness part.

---

> ### Author Response · Authors · 2025-11-24
> **Author Response**
>
> We thank the reviewer for the time and effort spent reviewing our paper. We are glad to see Decrypto described as _"elegantly designed"_ and as a _"clear, interpretable, and engaging framework"_ for theory of mind, competition and coordination. We also appreciate the recognition of our "extensive experiments" and of our attention to eliminating confounding factors, which was a key guiding principle for our work.
>
> We also thank the reviewer for their comment on the applicability of our benchmark to open-ended human interactions. We agree that Decrypto represents a simplified and artificial environment for theory of mind, and therefore does not fully capture all the nuance and ambiguity characteristic of freeform communication.
>
> This is intentional: the text- and turn-based structure is well suited to evaluate the current generation of AI agents, which are ill-equipped to handle interruptions, multiple concurrent speakers, and other non-verbal cues that are common in real-world communication. This aligns with Hu et al.'s call for _"species-fair"_ evaluation of ToM in LLMs, one that _"more directly measure[s] ToM while minimizing auxiliary demands"_ [1]. Indeed, in the same work, the authors argue that _"if our goal is to understand LLMs scientifically, then our tests should be 'pure', in the sense that they should isolate the targeted cognitive capability of interest"_, which was the objective of our work.
>
> Decrypto therefore evaluates ToM capabilities under controlled conditions: in a constrained environment that minimizes confounding factors and auxiliary demands, and with clearly defined roles and rules. Yet, even with such simplifications, current LLMs struggle with core ToM capabilities. This establishes a conservative upper bound on their abilities, with their performance under real-world scenarios potentially much worse.
>
> Finally, we also remark that our benchmark is nearly a 1-to-1 reimplementation of the award-winning board game on which it is based. So while Decrypto is an "artificial" assessment of ToM, it is not a toy one, but one that has adequate complexity to constitute an interesting (and fun) challenge for human players.
>
> We once again thank the reviewer for their time and hope our response clarifies the ecological validity and applicability of Decrypto as a ToM benchmark. If so, we hope they consider increasing their support for our paper.
>
> **Ref:**
>
> [1] Jennifer Hu, Felix Sosa, and Tomer Ullman. Re-evaluating theory of mind evaluation in large
> language models. arXiv preprint arXiv:2502.21098, 2025.

---

### Official Review · Reviewer_vmgS · 2025-11-03

**Soundness:** 2
**Presentation:** 1
**Contribution:** 1
**Rating:** 4
**Confidence:** 4

**Summary:**

DECRYPTO proposes a multi-agent, word-association game benchmark (inspired by the board game Decrypto) to test cooperation, competition, and theory of mind (ToM) in LLMs.  This work evaluates a range of open/closed models, adds specialist embedding baselines (GloVe/Word2Vec), and designs two interactive ToM experiments (representational change/false belief; perspective taking). They report that LLMs trail simple embedding baselines in cooperative play, excel as interceptors against other LLMs, and perform weakly on stronger ToM variants.

**Strengths:**

- an interesting ToM setting with a clean, language-only, interactive testbed for multi-agent reasoning and ToM that avoids many common confounds
- broad evaluations across model families/sizes; cooperative and competitive regimes; human-AI cross-play; prompt variants; also two adapted experiments (RC/FB and PT) provide diagnostic granularity

**Weaknesses:**

- section 4 feels overlong and under-integrated with ToM claims. Much of it re-states known concepts (zero-shot, OOD) without showing how these choices sharpen or test ToM hypotheses. Lines 190–209 are especially verbose, and the assertions that “specialists can overfit DECRYPTO” are plausible but not verified in this setting; also, how do authors ensure some foundation models do not see DECRYPTO in their training?
- the linkage between word-association mismatch and ToM is unclear. The paper attributes cross-play failures to “different word associations” but it’s not shown how this specifically implicates ToM rather than lexicon alignment. If ToM is the target construct, the paper should formalize ToM competence operationally (beyond game wins) and disentangle it from vocabulary grounding

- How do wins reflect ToM instead of public-knowledge heuristics? It remains possible models succeed by exploiting hint history or broad world knowledge rather than reasoning about another agent’s beliefs. The RC/FB/PT tasks help, but the main game results are not causally tied to ToM competence (no interventions that selectively alter others’ knowledge to see if Alice adapts).

- the benchmark is named after a game by others. The paper introduces a variant (three players, specific rules), claims RSA grounding, and an interactive ToM platform. The contribution would read cleaner if the name distinguished the original game from the benchmark extensions here. Also what is RSA formalization is used for?

- results discussion (sec. 5) is often post-hoc and shallow. Coordination/competition subsections describe outcomes but rarely test why (e.g., ablations that manipulate Eve’s access). Several insightful observations (e.g., o3-high overestimates interception rate) are stated without follow-ups.

- ToM experiments (Sec. 5.1) could be structured more cleanly; that is, the two procedures and findings are interwoven, and it is recommended to move the PT and RC/FB flow diagrams (Fig. 5/6) into the main text (instead of using heavy text for a better clarity) and present each experiment as Protocol → Metrics → Results → Takeaway would improve clarity.

- some claims need tightening. In lines 244–245, shared embeddings may yield shared similarity structure, not necessarily ToM. Any empirical support? Also, can authors justify “Tweak K to operate in a regime …” (lines 249–250)?

-  Sec. 3 (benchmark description) repeats mechanics and metrics (miscommunication/intercept) more than needed; the “future-proof” subsection collects heterogeneous points. Consider tightening and moving important, frequently referenced appendices (L, M; ToM diagrams) into the main text.

- Some citations (e.g., “Hu et al.” at line 43) lack full details; figures 5, 6 and sections L, M (critical to ToM) sit in the appendix, but intensively discussed in main text

**Questions:**

- How do you empirically separate belief modelling from shared word associations?

- Can you run interventions that alter Eve’s accessible history (hide a hint; inject misleading public info) and check whether Alice systematically adapts hints? That would more directly tie wins/losses to ToM.

- For the PT finding where many models predict near-certain interception on turn 1: did you verify whether they use that prediction to change hint strategy? If not, why is PT not feeding back into decisions? Can authors elaborate more on the failure: "that of integrating ToM reasoning in decision-making", how ToM is integrated in decision-making


- Can you demonstrate overfitting with a specialist agent?
- which concrete design choices (hint constraints, history visibility, metrics) follow from RSA predictions?

---

> ### Author Response · Authors · 2025-11-24
> **Author Response (1/2)**
>
> We thank the reviewer for their detailed feedback, including on the structure of our paper. We are glad that the reviewer liked our _"clean, language-only, interactive testbed"_ for ToM and multi-agent reasoning, as well as our _"broad evaluations"_, which provide _"diagnostic granularity"_. We respond to the reviewer's concerns below.
>
>
> # Content:
>
> ## 1. Why Decrypto requires ToM and connection to RSA
>
> We thank the reviewer for questions on this topic. As written on line 120, we "explicitly [show] that
> agents must model each other’s beliefs and perform second-order ToM for optimal play." This follows directly from formalizing Decrypto as a pragmatic inference game under the Rational Speech Act [1]. We present this derivation in Appendix H and refer to it in our response.
>
> > _"How do wins reflect ToM instead of public-knowledge heuristics?"_
>
> The RSA framework already covers this generally, and we discuss it in the context of Decrypto starting from eq.1: public knowledge establishes which hints and which keywords are semantically compatible (e.g. `fusion` is compatible with the keywords `jazz` and `star`). However, decoding the intended meaning requires reasoning about both literal interpretations and a model of Alice (eq. 7), which includes Alice's own model of Eve. The last paragraphs before Appendix H.1 also show what happens if the Decoder's mental model is imperfect and Appendix H.2 discusses the case where the players' knowledge differs, as is the case for models with different training cutoffs.
>
> Similarly, Appendix H.1 shows that Alice's likelihood of getting intercepted directly depends on how well she models Eve's guess probabilities.
>
> > _"Game results are not causally tied to ToM competence"_
>
> The aforementioned results mathematically tie wins/losses, and more specifically miscomms and intercepts, to failures of modelling other players. Our results in Section 5.1 also provide a clear diagnostic for why interceptions are so common in gameplay: _"LLMs do not attempt to model other agents before making decisions, and also struggle to model them if explicitly asked to."_ (End of Section 5)
>
> Finally, we included a puzzle in Figure 1, with the goal ​of _"[helping] readers intuit the role of ToM in Decrypto."_ This is why, on line 122, we recommended attempting the puzzle and reading the rationale provided in Appendix C.
>
> > _"which design choices follow from RSA predictions?"_
>
> None. We designed the benchmark to very closely match the original boardgame. This ensures its complexity and relevance for human-AI evaluations. The RSA formalism came after, but Decrypto can be described as a pragmatic inference game with mild assumptions, as we do in Appendix H.
>
> ## 2. On baselines and specialist agents
>
> > _"Can you demonstrate overfitting with a specialist agent?"_
>
> Yes, we already do so in Figure 2. We hardcoded two word embedding baseline agents to play Decrypto. They can play at an arbitrary level, but can do nothing else, and thus represent the most aggressive form of overfitting to a task.
>
> Furthermore, those baselines have a tunable parameter, $K$, which determines the size of the vocabulary per keyword from which Alice draw hints. By setting $K$ to a very high value, Alice can sample arbitrarily dissimilar hints, while still ensuring perfect communication with Bob, as long as Bob shares the exact same word embeddings and algorithm. Thus, at high $K$ value, the agent is also overfit to its partner, with catastrophically many miscommunications when paired with a different partner.
>
> > _"can authors justify 'Tweak K to operate in a regime …' "_
>
> Certainly. On line 156, we describe how we are interested in ad-hoc coordination with unseen partners, not merely in "solving" the game with a perfect win rate. Thus, when comparing LLMs to the baselines, we set $K=16$. In Figure 2, this roughly corresponds to the value beyond which the baseline-LLM miscommunication rate starts to increase significantly.
>
> We clarify this in the updated paper.
>
> > _"shared embeddings may yield shared similarity structure, not necessarily ToM"_
>
> Shared embeddings and inter-compatible Encoder/Decoder algorithms mean that the Decoder can perfectly recover the distribution of possible Encoder hints for any code. Similarly, the Encoder chooses hints following a perfect model of how they will be interpreted, and with perfect rationality. Thus, the Encoder and Decoder can be said to share perfect ToM. The case where the Encoder and Decoder have different embeddings also maps to the uncertain RSA framework, which we describe on line 1048 of the updated paper.

---

> > ### Author Response · Authors · 2025-11-24
> > **Author Response (2/2)**
> >
> > # Presentation:
> >
> > > _"the benchmark is named after a game"_
> >
> > Naming benchmarks after their eponymous games is not uncommon, with Hanabi [2] and Overcooked [3] being two highly cited examples. Our benchmark is nearly a 1:1 adaptation of the official 3-player variant of Decrypto and so we followed the convention established by those prior works.
> >
> > > _"present each experiment as Protocol → Metrics → Results → Takeaway"_
> >
> > This is good advice, and already do so, except for each ToM experiment independently. This was the result of feedback received in a previous review cycle, which recommended that the sub-section for each ToM experiment be self-contained, given the unusual protocol. We believe this is preferred by most readers, with Reviewer HFtN describing our presentation as _"clear and well-structured"_.
> >
> > > _"section 4 feels overlong and under-integrated with ToM claim"_
> >
> > Our benchmark is not limited to ToM; it also investigates cooperation and competition. Section 4 therefore describes the different solution concepts supported by Decrypto before introducing our baselines.
> >
> > > _"moving important, frequently referenced appendices (L, M; ToM diagrams) into the main text."_
> >
> > We thank the reviewer for the suggestion, which will improve the readability of our paper. We have now updated the paper and moved the ToM diagrams to the main text.
> >
> > Appendices L and M contain representative failure cases and model outputs, spanning a total of 8 pages. Meanwhile, the ICLR page limit for the main text is 10 pages. Can the reviewer please clarify what they meant?
> >
> > # Questions:
> >
> > > _"how do authors ensure some foundation models do not see DECRYPTO in their training?"_
> >
> > We make no such claims. Making the benchmark future-proof meant designing it such that if information about the Decrypto leaks into the training data, it is unlikely to significantly impact model performance. The interactive, combinatorial, open-ended and multi-agent aspects of the game all contribute to this end, as we describe in Section 3.
> >
> > In fact, many of the models tested can accurately output the rules of Decrypto when prompted, making their poor performance in the benchmark even more noteworthy.
> >
> >
> > > _"Can you run interventions that alter Eve’s accessible history"_
> >
> > All our results support the notion that Alice makes no attempt to model Eve during regular gameplay and that she incorrectly models Eve if explicitly prompted to model her. This is the root cause behind the majority of games ending in an interception. Thus, we believe that any interventions upon Eve's context or behaviour is unlikely to be noticed and taken advantage of by Alice.
> >
> > Therefore, can the reviewer therefore please clarify what intervention they had in mind why they would expect Alice to adapt at all, when she seldom even adapts to being intercepted?
> >
> > > _"why is PT not feeding back into decisions?"_
> >
> > We conducted experiments on a branching context, so it does not affect gameplay. Given the chance, models do change their hints following that prediction, but the new hints are equally likely to get intercepted. This is not surprising, since they have poor prediction skills to begin with.
> >
> > > _"how ToM is integrated in decision-making"_
> >
> > The decision here is the choice of hints. Normally, we would expect Alice to reason about Eve's knowledge before locking in her hints. This could be explicit (e.g. _"Let's think: what would Eve guess if..."_) or implicit in the model's activations. Regardless of the form it takes, the outcome of this process would be hints that Alice is reasonably confident will not get intercepted. What our PT results show is that Alice did not even attempt to reason about Eve when choosing her hints, since she subsequently predicts that Eve will intercept those very hints.
> >
> > We once again thank them for their detailed review. We look forward to any follow-up discussion and otherwise hope we have addressed all their concerns. If so, we invite them to revisit their review and increase their support for our submission.
> >
> > Kind regards,\
> > The Decrypto Authors
> >
> > **Ref:**
> >
> > [1] Noah D Goodman and Michael C Frank. Pragmatic language interpretation as probabilistic inference, 2016.
> >
> > [2] Bard et al., The Hanabi Challenge: A New Frontier for AI Research, 2019
> >
> > [3] Carroll, Micah, et al., On the utility of learning about humans for human-ai coordination. 2019

---

### Author Response · Authors · 2025-12-01
**Additional Clarifications for the AC**

Dear Area Chair,

Given the recent emergency changes to the review process, we understand that ACs must make their decisions this year with limited reviewer follow-up. In our case, none of the reviewers responded to our rebuttal before comments were disabled, so the available context is limited to the original reviews, our rebuttal, and the revised manuscript.

If there is any additional information, clarification, or feedback that would help you in making the final decision, we would be happy to address it before the deadline for the final author response.

Kind regards,
The Decrypto Authors

---

### Meta-Review · Area_Chair_KQm6 · 2026-01-12

**Summary:**

This paper introduces Decrypto, an interactive, language-only, game-based benchmark designed to evaluate multi-agent reasoning and Theory of Mind (ToM) in large language models. Inspired by the board game Decrypto, the benchmark isolates pragmatic inference and belief modeling by minimizing confounding factors such as symbolic or spatial reasoning. The authors present extensive empirical evaluations across open- and closed-source LLMs, human–AI cross-play, and adaptations of classic cognitive science tasks (e.g., false belief and perspective taking). The results consistently show that current LLMs lag behind humans and even simple embedding-based baselines, suggesting persistent deficiencies in ToM reasoning.

Reviewers’ concerns focused on whether Decrypto cleanly measures Theory of Mind versus confounds such as shared lexical associations, public-knowledge heuristics, or coordination artifacts; several felt the main gameplay metrics (wins/miscomms/intercepts) are not causally tied strongly enough to belief modeling without stronger interventions. Reviewers also noted that, while the experiments are broad, the analysis can be overly descriptive and lacks deeper causal/mechanistic explanations of model failures. Finally, there were presentation/positioning issues (verbosity, key diagrams/definitions in appendices, and an insufficiently explicit link between the RSA framing and concrete benchmark design choices).

**Reviewer Concerns:**

The decision is driven by borderline but unresolved concerns about conceptual grounding, causal interpretation, and clarity of contribution, rather than by issues of correctness or experimental rigor.

A central concern is whether performance in Decrypto cleanly measures Theory of Mind, as opposed to related but simpler factors such as shared lexical associations, public-knowledge heuristics, or coordination biases. While the authors provide an RSA-based formalization and auxiliary ToM experiments, several reviewers felt that the main game outcomes (wins, miscommunications, interceptions) are not causally tied strongly enough to belief modeling, and that additional interventions (e.g., selectively altering agents’ knowledge or access) would be needed to more decisively isolate ToM reasoning.

Reviewers also questioned the depth of insight provided by the empirical analysis. Although the experiments are broad, parts of the results discussion remain descriptive and post-hoc, with limited mechanistic or causal analysis explaining why particular models fail or succeed in specific scenarios. As a result, the benchmark’s diagnostic power, while promising, was seen as not yet fully realized.

Finally, there were presentation and positioning concerns. Some reviewers found the paper overly verbose, with key ToM-related diagrams and definitions placed in appendices, and noted that the connection between the benchmark, RSA theory, and concrete design choices could be made more explicit. While the rebuttal addressed many of these points (including moving diagrams and clarifying claims), these improvements do not fully resolve the perception that the contribution would benefit from further tightening and refocusing.

**Reviewer Scores:**

- Reviewer vmgS: Likely no change. While the rebuttal addressed several clarification and presentation issues, the reviewer’s core concerns about whether Decrypto cleanly isolates Theory of Mind (vs. lexical alignment or public-knowledge heuristics) would likely remain.
- Reviewer fmHC: Likely no change. The rebuttal clarified scope and intent, but did not add the deeper causal or mechanistic analyses the reviewer was looking for, so their marginally positive score would likely remain borderline.
- Reviewer E5CK: Likely no change. The rebuttal addressed missing citations and clarified positioning relative to prior benchmarks, but concerns about novelty and the limited richness of multi-agent dynamics would likely persist.
- Reviewer HFtN: Possibly a slight increase. The rebuttal added clarifications, examples, and discussion that directly respond to several of this reviewer’s questions, but the score would likely remain near-threshold rather than strongly positive.

---

### Decision · Program_Chairs · 2026-01-26

Reject